# Probabilistic Transformer:
# Modelling Ambiguities and Distributions
# for RNA Folding and Molecule Design

**Jörg K.H. Franke**[1], **Frederic Runge**[1], **Frank Hutter**[1,2]
[1]Department of Computer Science, University of Freiburg, Germany
[2]Bosch Center for Artificial Intelligence, Renningen, Germany
`{frankej,runget,fh}@cs.uni-freiburg.de`

## Abstract

Our world is ambiguous and this is reflected in the data we use to train our algorithms. This is particularly true when we try to model natural processes where collected data is affected by noisy measurements and differences in measurement techniques. Sometimes, the process itself is ambiguous, such as in the case of RNA folding, where the same nucleotide sequence can fold into different structures. This suggests that a predictive model should have similar probabilistic characteristics to match the data it models. Therefore, we propose a hierarchical latent distribution to enhance one of the most successful deep learning models, the Transformer, to accommodate ambiguities and data distributions. We show the benefits of our approach (1) on a synthetic task that captures the ability to learn a hidden data distribution, (2) with state-of-the-art results in RNA folding that reveal advantages on highly ambiguous data, and (3) demonstrating its generative capabilities on property-based molecule design by implicitly learning the underlying distributions and outperforming existing work.

## 1 Introduction

Transformer models [1] are the architecture of choice for many applications. Next to a wide range of NLP applications, such as language modelling [2, 3, 4] and machine translation [1, 5], they are also very effective in other disciplines, such as computer vision [6, 7], biology [8, 9], and chemistry [10, 11]. An additional challenging application for which transformers are promising is RNA folding, where the goal is to model a secondary structure (represented in dot-bracket notation [12]) based on a given sequence of nucleotides. RNA data is highly ambiguous since it is collected with different techniques, resolutions, protocols, and even contains natural ambiguities since there exist multiple structures for the same RNA sequence in the cell [13, 14] and the same structure can be caused by multiple sequences. Similar to RNA structures, molecules can be represented as sequences by using the *simple molecular line entry system* (SMILES) [15] and transformers arise as the architecture of choice in the flourishing field of molecule design [11, 16].

Although Transformers have outstanding performance, the deterministic core of the architecture could harm performance in real-world applications like RNA folding. If collected data contains noisy labels or ambiguous samples, a vanilla Transformer model can only express uncertainties in the softmax output but not in the latent space. When sampling, sequential interdependencies can only be modelled in a decoder setting but not in an encoder-only setting. We address these limitations by proposing a Probabilistic Transformer[1] (ProbTransformer) that models a hierarchical latent distribution and performs sampling in the latent space. The ProbTransformer can represent ambiguities in these

---

[1]Source code is available at github.com/automl/ProbTransformer

distributions and refine the sampled latent vectors within the computational graph. This is in line with recent findings in cognitive science that suggest that the human brain both represents probability distributions and performs probabilistic inference [17, 18]. Our approach is based on the idea of combining the transformer architecture with a conditional variational auto-encoder (cVAE) [19], but in a hierarchical fashion similar to the hierarchical probabilistic U-Net [20]. Therefore, we introduce a new probabilistic layer and incorporate it after the attention and feed-forward layer (Section 3). In this way, we preserve the global receptive field through the attention mechanism and remain independent of other enhancements in the transformer ecosystem [21]. To train the latent distributions, we make use of the generalized evidence lower bound (ELBO) with constrained optimization (GECO) [22] and introduce an annealing technique to adapt a hyperparameter online.

We see our contributions in three aspects:

- The introduction of the ProbTransformer, a novel hierarchical probabilistic architecture enhancement to the Transformer ecosystem.
- Our training procedure using GECO, the analysis of the sensitivity of its hyperparameter $\kappa$, and the introduction of the online adaption technique *kappa annealing* which could be beneficial for variational training with ELBO in general.
- A comprehensive empirical analysis that verifies the ProbTransformer's capability to learn and recover data distributions on a novel synthetic sequential distribution task, assesses its capability of handling data ambiguities in practice by achieving state-of-the-art performance in RNA folding, and demonstrates its generative character by outperforming existing work in Molecule Design.

We first clarify notation and recap the cVAE [19] (Section 2), and then introduce the ProbTransformer (Section 3). We then present our experiments on the sequential distribution task (Section 4.1), RNA folding (Section 4.2), Molecule Design (Section 4.3), as well as our ablation study (Section 4.4), discuss related work (Section 5) and conclude (Section 6).

## 2   Background

**Transformer Notation**   The Transformer [1] is a self-attention based sequence-to-sequence model introduced as an encoder-decoder architecture. However, both encoder-only and decoder-only versions are very successful by themselves [2, 23], and in our work, we focus on either of these two versions. An encoder or decoder has $N$ blocks and each block consists of a multi-head (masked) attention followed by a feed-forward layer. Around both of these layers there are residual connections [24] followed by layer normalization [25]. We use the parameterization with $D_{model}$ dimensions in the residuals and attention, $H$ heads per attention, $D_{ff}$ latent dimensions in the feed-forward layer, and $N$ for the blocks in the Transformer. In addition, we define $X \widehat{=} (x_1, \ldots, x_S)$ as the input sequence of length $S$, $Y \widehat{=} (y_1, \ldots, y_S)$ as the target sequence, and $\hat{Y} \widehat{=} (\hat{y}_1, \ldots, \hat{y}_S)$ as the predicted sequence.

**Conditional Variational Auto-Encoder (cVAE)**   The cVAE [19] is a deep conditional generative model and an extension of the Variational Auto-Encoder [26, 27]. During inference, the cVAE aims to generate a distribution for an output $\mathbf{y}$ conditional on an input $\mathbf{x}$. More specifically, given an input $\mathbf{x}$, a latent variable $\mathbf{z}$ is drawn from a (conditional) prior model $p_\theta(\mathbf{z}|\mathbf{x})$ and used as an additional input to the generation model $p_\rho(\mathbf{y}|\mathbf{x}, \mathbf{z})$ in order to generate the prediction $\hat{\mathbf{y}}$. The cVAE is trained by minimizing the negative evidence lower bound (ELBO), $\mathcal{L}_{ELBO}$:

$$\mathcal{L}_{ELBO}(\mathbf{x}, \mathbf{y}; \theta, \rho, \psi) = \mathbb{E}_{\mathbf{z}^{post} \sim q}(-\log p_\rho(\mathbf{y}|\mathbf{x}, \mathbf{z}^{post})) + \mathrm{D_{KL}}(q_\psi(\mathbf{z}|\mathbf{x}, \mathbf{y}) \parallel p_\theta(\mathbf{z}|\mathbf{x})) \qquad , \quad (1)$$

where $q_\psi(\mathbf{z}|\mathbf{x}, \mathbf{y})$ describes the posterior model (called "recognition network" in [19]). All models are neural networks, and the prior and posterior models each output a mean and variance of a Gaussian distribution which represents the distribution of the latent $\mathbf{z}$. During training, $\mathbf{z}^{post} \sim q_\psi(\mathbf{z}|\mathbf{x}, \mathbf{y})$ is sampled from this posterior model and used as input to the generation model $p_\rho(\mathbf{y}|\mathbf{x}, \hat{\mathbf{z}})$, whose output $\hat{\mathbf{y}}$ is compared to the ground true with a cross-entropy loss. This objective at training time can be viewed as a reconstruction task due to the target sequence input to the posterior, which is an easier task than prediction. The Kullback-Leibler (KL) divergence term $\mathrm{D_{KL}}(\cdot \parallel \cdot)$ aims to align the (conditional) prior model $p_\theta(\mathbf{z}|\mathbf{x})$ and the posterior model $q_\psi(\mathbf{z}|\mathbf{x}, \mathbf{y})$. The two losses are added and then used to train the prior, posterior, and generation models jointly, employing the reparameterization trick [28].

# 3 Probabilistic Transformer

Building on concepts of the cVAE, we enhance the Transformer model to a probabilistic Transformer (ProbTransformer) by adding a probabilistic feed-forward layers (prob layer) to $M \in \{1, \ldots, N\}$ of the existing blocks. We introduce our approach for an encoder-only model to simplify notation. It also applies to the decoder-only model but not directly to an encoder-decoder model because our training setup requires the alignment of the source and target sequences in the posterior input, see Section 3.2. Whether all or only a selection of blocks are enhanced with prob layers is a design decision and examined empirically in Section 4.4. In a block, we place the prob layer after the attention and feed-forward layer. The prob layer parameterizes a multivariate Gaussian distribution $\mathcal{N}(\boldsymbol{\mu}, \Sigma)$ with a diagonal covariance $\Sigma = \boldsymbol{\sigma}^2 \mathbf{I}$, where $\mathbf{I}$ is the identity matrix, used to sample a latent vector $\mathbf{z} \in \mathbb{R}^{D_z}$ for each position in the sequence. We denote by $X$ and $Y$ the input and target sequences of length $S$, by $Z_m \widehat{=} (\mathbf{z}_{m,1}, \ldots, \mathbf{z}_{m,S})$ the position-wise sampled sequence of latent distributions in block $m$, by $\mathbf{Z} \widehat{=} (Z_1, \ldots, Z_M)$ all sequences of latent distributions of all blocks, and by $\mathbf{Z}_{<m} \widehat{=} (Z_1, \ldots, Z_{m-1})$ all sequences of latent distributions of all blocks before $m$. At inference time, we sample $Z_m$ and add it to the computation graph of the current block, similar to sampling from the (conditional) prior in the cVAE. However, in contrast to the cVAE, our sampling is conditioned hierarchically; the latent realization $\mathbf{z}_{m,s}$ at block $m$ and sequence position $s$ depends on all previously sampled latent vectors of any position in previous Transformer blocks due to the Transformer architecture:

$$\mathbf{z}_{m,s} \sim p(\mathbf{z}_{m,s}|\mathbf{Z}_{<m}, X). \tag{2}$$

A sequential relation between the positional distributions is achieved by the attention mechanism: while the samples $\mathbf{z}_{m,s}$ are drawn independently for each position $s \in \{1, \ldots, S\}$, the following attention operations relate them and turn the position-wise samples into a joint distribution over sequences refined in higher blocks.

As a result of the hierarchical composition of the ProbTransformer and the sequential relation, prior model $p_\theta(Z_m|\mathbf{Z}_{<m}, X)$ and the generation model $p_\rho(Y|X, \mathbf{Z})$ effectively become a single *predictive model* $P_\phi(Y|X)$, which differs from the modeling strategy of the cVAE. This model approximately marginalizes over latent variables $\mathbf{Z}$: $P_\phi(Y|X) = \int_\mathbf{Z} p_\phi(\mathbf{Z}|X) \times p_\phi(Y|X, \mathbf{Z}) \, d\mathbf{Z}$, and we can sample from it by hierarchically sampling $p_\phi(\mathbf{Z}|X)$ decomposed as

$$p_\phi(\mathbf{Z}|X) = p_\phi(Z_M|\mathbf{Z}_{<M}, X) \cdot \ldots \cdot p_\phi(Z_0|X), \tag{3}$$

followed by sampling from $p_\phi(Y|X, \mathbf{Z})$. At inference time, we can sample different predictions from the predictive model $\hat{Y} = P_\phi(Y|X)$.

However, we can also use the mean of the respective (Gaussian) distributions instead of sampling from them. We denote this as *mean inference* in contrast to *sample inference*. In the following, we introduce the prob layer architecture in detail before explaining our training setup and the learning objective.

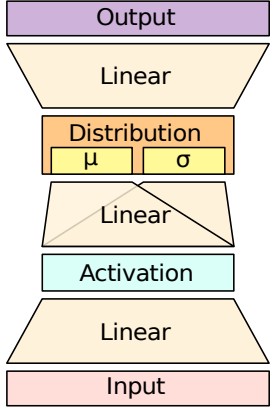

Figure 1: Probabilistic Feed-Forward layer.

## 3.1 Position-wise Probabilistic Feed-Forward Network

Similar to the feed-forward layer, the prob layer is applied to the latent representation of each position $s \in \{1, \ldots, S\}$ of the input sequence $X$ independently. Figure 1 provides a visual description. It consists of a linear layer that transforms from the model dimension $D_{model}$ to the distribution dimension $D_z$ of the probabilistic latent space, followed by an activation function. Further, two linear layers with $D_z \times D_z$ weight matrices generate the latent representation of a conditional Gaussian distribution with mean $\boldsymbol{\mu}_{m,s} \in \mathbb{R}^{D_z}$ and log variance[2] $\log \boldsymbol{\sigma}_{m,s}^2 \in \mathbb{R}^{D_z}$

---

[2]We use $\log \boldsymbol{\sigma}_{m,s}^2$ to avoid negative variance values and for numerical stability. We obtain the variance $\boldsymbol{\sigma}_{m,s}^2$ by $\boldsymbol{\sigma}_{m,s}^2 = e^{\log \boldsymbol{\sigma}_{m,s}^2}$.

for the prob layer in block $m$ at position $s$:

$$\boldsymbol{\mu}_{m,s} = \text{Linear}_{\mu,m}(\text{Act}(\text{Linear}_{\text{In},m}(\mathbf{x}_{m,s}))) \tag{4}$$

$$\log \boldsymbol{\sigma}_{m,s}^2 = \text{Linear}_{\sigma,m}(\text{Act}(\text{Linear}_{\text{In},m}(\mathbf{x}_{m,s}))) \tag{5}$$

$$\mathbf{z}_{m,s} \sim \mathcal{N}(\boldsymbol{\mu}_{m,s}, \boldsymbol{\sigma}_{m,s}^2 \mathbf{I}) =: p(\mathbf{z}_{m,s}|\mathbf{Z}_{<m}, X) \tag{6}$$

$$\mathbf{y}_{m,s} = \text{Linear}_{\text{Out},m}(\mathbf{z}_{m,s}) \tag{7}$$

As mentioned before, during *sample inference*, we can sample from the distribution (Equation 6) and during *mean inference* we use $\mathbf{z}_{m,s}\widehat{=}\boldsymbol{\mu}_{m,s}$. An additional linear layer is used to compute the layer's output $\mathbf{y}_{m,s} \in \mathbb{R}^{D_{model}}$ (Equation 7). Similar to the attention and feed-forward layer, we employ a residual connection [24] followed by a layer normalization [25].

### 3.2 Training Setup and Learning Objective

We optimize the ELBO as the standard practice for conditional training [19]. This requires a variational posterior $Q_\psi(\mathbf{Z}|X, Y)$ that depends on both the input sequence $X$ and target sequence $Y$. We model this posterior with a separate ProbTransformer with the same architecture as the predictive model except for an additional input embedding for the target sequence $Y$ and no output generation layers, since we are only interested in the latent sampling $\mathbf{Z}^{post}$ and not in the actual model output. During training, we first run the posterior model and sample the latent $\mathbf{Z}^{post} \sim Q_\psi$. In a second step, we run the predictive model but use the latent realization $Z_m^{post}$ instead of sampling an own $Z_m$ in each prob layer $m$. Figure 2 illustrates this training setup.

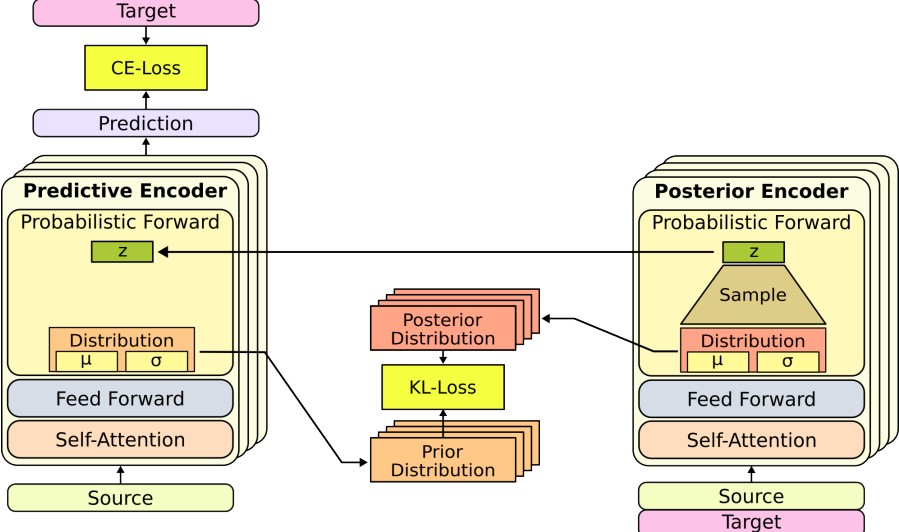

Figure 2: Training setup of the ProbTransformer: The predictive and posterior encoder trained jointly.

The negative ELBO loss $\mathcal{L}_{ELBO}$ (Equation 1) is composed of a reconstruction loss $\mathcal{L}_{rec}$ and a KL divergence $D_{KL}$ between the prior distribution $P_\phi$ conditioned on the latent $\mathbf{Z}^{post}$ and the posterior distribution $Q_\psi$. The reconstruction loss is the cross-entropy between the predicted sequence $\hat{Y}$ and the true sequence $Y$ while using the latent $\mathbf{Z}^{post}$ from the posterior model:

$$\mathcal{L}_{rec} = \mathbb{E}_{\mathbf{Z}^{post} \sim Q}\big(-\log p_\phi(\mathbf{y}|X, \mathbf{Z}^{post})\big), \tag{8}$$

and $D_{KL}$ is the sum of KL divergences between the hierarchically decomposed distributions $P$ and $Q_\psi$ at each prob layer $m$ for all positions $s$:

$$\mathcal{D}_{KL} = \frac{1}{S} \sum_{s=1}^{S} \sum_{m=1}^{M} D_{\text{KL}}\big(q_\psi(\mathbf{z}_{m,s}^{post}|\mathbf{Z}_{<m}^{post}, X, Y) \,\|\, p_\phi(\mathbf{z}_{m,s}|\mathbf{Z}_{<m}^{post}, X)\big) \tag{9}$$

With the default $\mathcal{L}_{ELBO}$ loss, we discovered instability in the training and performance issues, even with a weighting factor $\beta$ [29]. Therefore, we follow [20] to avoid convergence issues with the classic

ELBO objective and make use of the Generalized ELBO with Constrained Optimization (GECO) objective [22], optimizing the Lagrangian

$$\mathcal{L}_{GECO} = \lambda(\mathcal{L}_{rec} - \kappa) + \mathcal{D}_{KL}. \tag{10}$$

The Lagrange multiplier $\lambda$ balances the terms and is initialized to 1, added to the learnable parameter space and updated as a function of the exponential moving average of the reconstruction loss [20]. In the beginning of the training, there is high pressure on the reconstruction loss due to an increasing $\lambda$. Once the desired reconstruction loss $\mathcal{L}_{rec} \approx \kappa$ is reached, the pressure moves to the KL-term. Please find more information about the training dynamics in Appendix A.

$\kappa$ was set to a constant in the original work of [22], but this is problematic since it is a sensitive hyperparameter due to the training dynamics. Specifically, if $\kappa$ is chosen *too small*, possible failure modes include that $\mathcal{L}_{rec}$ never reaches it (with the pressure staying only on the reconstruction loss), that $\kappa$ will be reached by over-fitting, or that the posterior distribution collapses which leads to high $D_{KL}(Q_\psi \parallel P_\phi)$ and destabilizes the training. On the other hand, if $\kappa$ is chosen *too large*, the model can underfit or the reconstruction loss drops below $\kappa$, leading to a negative loss value and harming the training success significantly. To address this issue we introduce **kappa annealing** and adjust $\kappa$ during training. Let $\mathbf{L}_c = \frac{1}{K}\sum_{k=0}^{K} \mathcal{L}_{rec}(X_k, Y_k) - \kappa$ be the mean constrained difference for $K$ training samples in one epoch of training or a defined number of update steps. In our experiments, we find that initializing $\kappa$ slightly larger than the optimal CE loss value and updating it every epoch with

$$\kappa = \begin{cases} \kappa + \mathbf{L}_c & \text{if } \mathbf{L}_c < 0 \text{ and } \lambda \leq 1 \\ \kappa & \text{otherwise} \end{cases} \tag{11}$$

leads to more stable training behaviour, improved final performance, and reduces the need for expensive tuning of $\kappa$.

## 4 Experiments

We demonstrate the benefits of the ProbTransformer in three experiments: (1) using a novel synthetic sequential distribution task, we show the advantages of distributions in the latent space over the standard softmax output distribution of a vanilla Transformer. (2) We show the benefits of the ProbTransformer in dealing with ambiguous data in RNA folding. (3) We use the ProbTransformer to improve property-based Molecule Design by sampling from the latent space instead of a softmax-output. In an additional ablation study, we provide insights on the impact of kappa annealing and hierarchical prob layers.

### 4.1 Synthetic Sequential Distribution Task

In real-world generation tasks, the true distribution is not available and is only implicitly accessible by a dataset. To provide insights on the quality of distribution learning, we created a synthetic sequential distribution task and compare the ProbTransformer to two sampling methods of a vanilla Transformer.

**Data** We design the task to map a sequence of tokens from a source vocabulary $x \in \mathcal{V}_i^*$ to a sequence of target tokens from a target vocabulary $y \in \mathcal{V}_o^*$ with the same length. The tokens in the source sequence are used to build 'phrases" $\mathbb{P}$. Each phrase exists of $l$ tokens sampled with replacement (similar to the combination of words to phrases in a sentence). For each source token in each phrase, we randomly generate a unique distribution $p(y|x, \mathbb{P})$ over the target tokens depending on the current phrase. Further, we design the distribution sparsely so that no more than $k$ tokens from the target vocabulary have a non-zero probability. The training data consists of source and target sequence pairs with input sequences sampled with replacement from all phrases and target sequences sampled from the corresponding distributions.

**Setup** We use an encoder-only ProbTransformer model and enhance each block with a prob layer. We configure the task and the model to run on one GPU within few hours. Please find more information about the task and the configuration in Appendix B. For the inference of the vanilla Transformer we use MC dropout (using dropout during inference time, based on [30]) and sampling from the softmax distribution. For the inference of the ProbTransformer we sample from the predictive model and use the token with the highest output value. We generate 50 realizations per sample to create a predictive distribution of the target vocabulary.

**Results** We measure the generative performance of a model or sampling method with two metrics: The *Validity* describes the percentage of predicted tokens whose probability in the true distribution is not zero. Second, we measure the *KL-divergence* between the sampled distribution and the true distribution since sampling should reproduce the true target token distribution. As shown in Table 1, the ProbTransformer outper-

Table 1: The mean measures of five random seeds in the synthetic task.

| Model | | Validity | KL-div. |
|---|---|---|---|
| ProbTransformer | | **0.99** | **0.52** |
| Transformer | dropout | 0.93 | 12.71 |
| | softmax | 0.73 | 7.84 |

forms the MC dropout and softmax sampling, demonstrating its strong performance in probabilistic sequence modelling. Please note that this task does not consider interdependence sampling, where the sampling of one token in a sequence depends on the realization of others, while the next task on RNA folding does.

## 4.2 RNA Folding

An RNA's structure influences its function drastically [31] and the functional importance of RNA arguably is on par with that of proteins [32]. Since cellular RNAs typically have extensive secondary structures but limited tertiary structures [33], RNA folding is typically modeled as a function $\mathcal{F} : \Upsilon^* \rightarrow \Gamma^*$, where $\Upsilon$ and $\Gamma$ denote sequence and secondary structure alphabets, respectively. However, this is a simplified view on the RNA folding process, which ignores the fact that RNAs alter their structures dynamically, resulting in an ensemble of structures occurring with different probabilities [13, 14]. Further, real-world applications often require structure analysis of very similar sequences [34], sometimes folding into the same secondary structure. These ambiguities can hardly be captured with common approaches and ambiguous data is often removed for a better training result [35, 36, 37]. We address these issues with probabilistic modelling using our ProbTransformer and by keeping ambiguous training data, while implicitly measuring overfitting by explicitly removing similarity to the test data.

**Data** We collect a large data pool of publicly available datasets from recent publications [38, 39, 37, 40, 41] and split the predefined validation and test sets, VL0 and TS0 [37], from the pool. We derived a separate testset, *TSsameStruc*, from 149 samples of TS0 that share the same structure with at least one other sequence in TS0 and uniformly sampled a disjoint set of 20 sequences with more than one annotated secondary structure from the remaining pool to produce an ambiguous testset, *TSsameSeq*. Samples without pairs and with a sequence similarity greater than 80% to the test and validation sets were removed from the training pool. However, in contrast to previous work [37, 39, 42], we kept all remaining samples for training to capture ambiguities and the influence of small sequence changes on the structures. The final data consists of 52007 training and 1299 validation samples, and 1304 samples in the testset TS0 and 46 samples in the testset TSsameSeq. We observe 48092, 1304, and 20 unique sequences with up to 7 different structures for a single sequence, and 27179, 1204, and 46 unique structures with up to 582 different sequences for a single structure in the training, TS0, and TSsameSeq sets, respectively. This indicates that many sequences map to the same structure and different structures to the same sequence. We refer to Appendix C.1 for more information about the data.

**Setup** We use a 6-block encoder-only architecture with $D_{model}$ of $512$ for the vanilla Transformer and ProbTransformer. We train them for 200 epochs or 1M training steps. We compare them against the state-of-the-art deep learning algorithms, *SPOT-RNA* [37], MXFold2 [39], and a commonly-used dynamic programming approach, *RNAfold* [43] on the testset TS0 and its subset TSsameStruc using mean inference. To analyse the capabilities of the ProbTransformer to reconstruct structure distributions, we also infer the model 5, 10, 20, 50, and 100 times using sample inference on TSsameSeq, and compare against three commonly-used algorithms specialized on the prediction of structure ensembles via sampling from the Boltzmann distribution based on experimentally derived thermodynamic parameters: *RNAsubopt* [43], *RNAshapes* [44], and *RNAstructure* [45]. In contrast to previous work [37, 41, 39, 42], we do not use ensembling nor limit the set of accepted base pairs or secondary structures for post-processing, but independently train a CNN head on the output of the ProbTransformer that maps the string representation to an adjacency matrix representing the structure for evaluations on TS0 and TSsameStruc. Although we are aware of the problems with exact evaluations via F1 Score due to the dynamic nature of RNA structures [46], we report F1 Score for reasons of comparability to previous work; however, we further report the number of solved structures

(the ultimate goal of the task), and the average Hamming distance to achieve a better measure of distance. For more details on the setup, CNN head, and metrics, we refer to Appendix C.2.

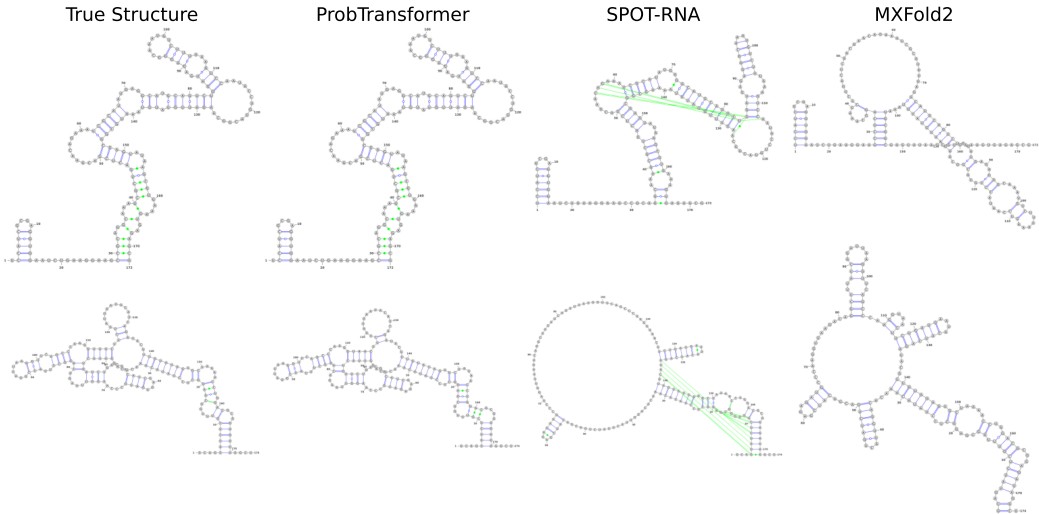

Figure 3: Example predictions of deep learning-based approaches for challenging RNAs from TS0. We show (top) a Group II catalytic intron (RF02001) and (bottom) a M-box riboswitch (RF00380).

Table 2: Structure fidelity of different RNA folding approaches on TS0 and TSsameStruc. For the ProbTransformer and the vanilla Transformer we show mean results of three random seeds.

| Model | TS0 | | | TSsameStruc | | |
|---|---|---|---|---|---|---|
| | F1 Score | Hamming | Solved | F1 Score | Hamming | Solved |
| ProbTransformer | **62.5** | **27.4** | **0.118** | **93.2** | **3.2** | **0.550** |
| Transformer | 50.5 | 35.3 | 0.084 | 89.5 | 4.6 | 0.481 |
| SPOT-RNA | 59.7 | 39.6 | 0.005 | 78.0 | 14.6 | 0.020 |
| MXFold2 | 55.0 | 42.1 | 0.014 | 74.6 | 17.1 | 0.067 |
| RNAFold | 49.2 | 48.0 | 0.008 | 59.2 | 25.5 | 0.020 |

**Results** Table 2 summarizes the results of all approaches on TS0 and TSsameStruc. Both attention-based approaches generally achieve strong results, but we observe that the ProbTransformer outperforms the vanilla Transformer across all measures on both sets. The ProbTransformer achieves the best performance in terms of F1 Score, solves more than eight times the structures compared to the next best approach (MXFold2) on both sets and achieves four times lower Hamming distance on TSsameStruc compared to SPOT-RNA, indicating that it has learned to handle ambiguous sequences. Our model is further capable of accurately reconstructing challenging structures, exemplarily shown for two structures that contain long-range interactions in Figure 3. A functional important class of base pairs are pseudoknots [47, 48], non-nested base pairs present in around 40% of RNAs [41] that are overrepresented in functional important regions [49, 48] and known to assist folding into 3D structures [50]. While RNAFold and MXFold2 cannot predict this kind of base pairs due to the underlying nearest neighbour model, Figure 3 as well as further results shown in Appendix C.3.2 suggests that the ProbTransformer can predict pseudoknots more accurately than SPOT-RNA if these are contained in the structures (Figure 9) and further rarely predicts pseudoknots if the structure is nested. However, a detailed analysis on the quality of pseudoknot predictions would be out of the scope of this work. When inferring the model multiple times on TSsameSeq, the ProbTransformer is the only model in our evaluation that could reconstruct more than one structure for a given sequence (for two sequences this was already achieved when inferring the model only five times). The average Hamming distances of the best predictions on TSsameSeq for every approach are summarized in Table 15 in Appendix C.3.2. The ProbTransformer improves the Hamming distance by up to 44%, indicating that the ProbTransformer reconstructs the overall structure ensemble very well. Additional results and example predictions for RNA folding are reported in Appendix C.3.

Table 3: Multi-property (TPSA+logP+SAS) conditional training on GuacaMol dataset (mean on five different seeds).

| Model | Validity | Unique | Novelty | TPSA MAD/SD | logP MAD/SD | SAS MAD/SD |
|-------|----------|--------|---------|-------------|-------------|------------|
| ProbTransformer | **0.981** | 0.821 | 1.0 | **2.47/2.04** | **0.22/0.18** | **0.16/0.14** |
| MolGPT | 0.973 | **0.969** | 1.0 | 3.79/4.80 | 0.27/0.35 | 0.18/0.26 |

## 4.3 Molecule Design

In the field of generative chemistry [51], deep generative models are employed to explore the chemical space [52, 53, 54]. However, biological applications typically require that the designed molecules have certain desired properties. For example, to penetrate the blood-brain barrier in order to interact with receptors of the nervous system, a molecule typically requires a certain permeability score [55, 56]. Conditional generation then refers to the problem of exploring the chemical space conditioned on molecule properties; a well-suited task to evaluate the ProbTransformer in a decoder-only setting against the state-of-the-art transformer decoder model, *MolGPT* [11].

**Data** We use the GuacaMol [57] training data and the evaluation protocol provided by [11]. For more information about the data we refer to Appendix D.1.

**Setup** We employ the same architecture as [11], enhanced with probabilistic layers, and train our model for 20 epochs or 300k training steps. During training, the model implicitly learns the properties of the training data by conditioning generation on molecular properties together with the input SMILES. At inference, novel molecules with multiple desired property values are generated by providing the model with a start token alongside with the desired property values while predicting the next token until either a maximum length is reached or the model produces an end-token. We condition the generation of molecules on three properties: the *synthetic accessibility score (SAS)*, the *partition coefficient (logP)*, and the *topological polar surface area (TPSA)*, using the same domains of values described in [11]. For each combination of property values, the model generates a total of 10000 molecules. Results are reported in terms of the mean average deviation (MAD) and the standard deviation (SD) relative to the range of the desired property values, as well as the *validity* of the generated compounds, their *uniqueness* in terms of the internal diversity of valid predictions, and their *novelty* compared to the training data. For more information about the measures and the general setup, we refer to Appendix D.2.

**Results** The results for the conditional generation of molecules are summarized in Table 3. We observe a high validity score and a novelty of 1.0, indicating that our ProbTransformer has learned the underlying SMILES grammar very well and does not suffer from overfitting to the train data. For the main task of conditional generation, the ProbTransformer clearly outperforms *MolGPT* across all measures, which highlights its ability to control multiple molecular properties during generation. The largest improvement can be observed for the SD and MAD scores of TPSA, an important measure for drug delivery in the body, improving the standard deviation (SD) by 57.5% and the mean average deviation (MAD) score by nearly 35%. We do not observe improvements in uniqueness compared to *MolGPT* but note that nearly perfect uniqueness could, e.g., be achieved by adding carbons *a posteriori* [58]. Additional results and prediction examples are shown in Appendix D.3.

## 4.4 Ablation Studies

We perform two ablation studies to provide more evidence for the usefulness of hierarchical probabilistic construction and kappa annealing. For this ablation, we use the RNA Folding task described in Section 4.2 with the same architecture and training steps. We provide more details in E.

**Hierarchical probabilistic design** Due to the possibility of having a prob layer in any block, we tested different setups from one prob layer (middle) up to all blocks enhanced with a prob layer. In Figure 4, we measure the performance improvement in Hamming distance compared to a vanilla Transformer on the TS0 and TSsameStruct sets. Performance saturates at 4 prob layers for the TS0 set but keeps improving on the TSsameStruct set. This is in line with our hypothesis that especially ambiguous samples profit more from the hierarchical architecture.

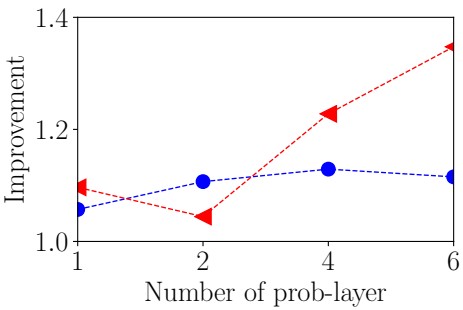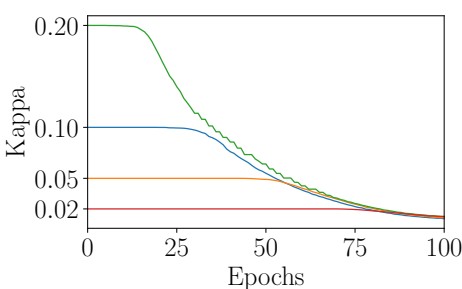

Figure 4: (Left) Performance improvement by number of prob layer: Dot (blue) on TS0 and triangle (red) on TSsameStruc. (Right) Kappa annealing with different initialization over 100 training epochs.

**Kappa Annealing** We assess the effect of kappa annealing when initializing $\kappa$ with different values. In this ablation, we compare a constant $\kappa$ to kappa annealing on the Hamming distance, see Table 4. In these results, kappa annealing always improves the performance and substantially stabilizes final performance across a range of initialization values. Figure 4 shows the adaption of $\kappa$ over a training run, demonstrating that it smoothly converges to a smaller value. Initializing with such a small value would yield much worse performance.

Table 4: Hamming distance with and w/o kappa annealing on different initialization.

| Kappa | w/o | annealing |
|-------|------|-----------|
| 0.02 | 29.4 | 29.2 |
| 0.05 | 28.0 | 28.0 |
| 0.1 | 29.4 | 27.9 |
| 0.2 | 32.2 | 29.2 |

## 5  Related Work

**Related Transformer Models** A series of works enhanced the encoder-decoder Transformer architecture with a distribution in the latent space between encoder and decoder. During inference, the model samples from these distributions. There is work in the field of music representation [59] which does not use a posterior model. Work in neural machine translation [60, 61] has a separate posterior encoder and enhances the ELBO objective. This is similar to the work of Lin et al. [62] which introduces a variational decoder layer for dialogue modelling and has a separate posterior encoder. Other work in text generation [63, 64, 65] uses the encoder for the predictive and posterior model. In contrast to our work, neither of the methods mentioned above model hierarchical distributions or use the GECO objective. The work of Pei et al. [66] introduces a hierarchical stochastic multi-head attention mechanism aiming at uncertainty estimation. In contrast, we preserve the original attention mechanism by adding a new layer. The work of Liu et al. [64] also introduces a $\beta$ scheduling in $\beta$-VAE ELBO objective [29] similar to our kappa annealing but focused on the KL loss instead of the reconstruction objective.

**Related work in RNA folding** Until recently, RNA folding was dominated by dynamic programming (DP) algorithms using either thermodynamic, statistical, or probabilistic scoring functions [67]. Learning-based approaches to the problem benefit from making few assumptions on the folding process and allowing previously unrecognized base pairing constraints [42], which recently led to state-of-the-art performance using deep learning [37, 42]. We discuss these state-of-the-art deep learning approaches in detail in Appendix C.4 and refer to a recent review [68].

**Related Work in Molecule design** While a plethora of deep learning-based models have been proposed for *de novo* generation of molecules in the past five years [69, 70, 71, 72, 73, 74, 75, 76, 77, 78, 79], only some methods yet approached the challenging task of generating molecules with (multiple) predefined property values (conditional generation). These include conditional RNNs [80], cVAEs [81], conditional adversarially regularized autoencoders [82], and more recently also Transformers [11]. Since [81] considers inorganic molecules and conditional generation is only performed in a case study in [82], and because of the usage of different evaluation protocols in [80] and [11], a correct evaluation and fair comparison of these approaches is out of the scope of this work. We, therefore, compare the ProbTransformer to the most similar and recent work in the field,

*MolGPT*, which, to our knowledge, is the current state-of-the-art in the field. For more details on the different methods, we refer to multiple excellent reviews [83, 84, 51, 53].

# 6 Discussion and Conclusion

We propose a novel probabilistic layer to enhance the transformer architecture with hierarchical latent distributions while keeping the global receptive field via the attention mechanism. The ProbTransformer samples interdependent sequences in one forward path. This sampling happens in the latent space, and the ProbTransformer can refine or interpret a sampled latent representation in a further layer. Compared to sampling from the softmax output distribution, this approach yields greater flexibility. It also is compatible with other enhancements to the Transformer model since it only adds a new layer but keeps everything else unchanged.

We showed the benefits of our approach in several experiments. To our knowledge, the ProbTransformer is the first learned RNA folding model that can provide multiple correct structure proposals for a given RNA sequence, which opens the doors to novel research paths in RNA structure prediction that are in line with experimental evidence for RNA structural dynamics from, e.g., NMR studies, such as fraying [85], bulge migration [86], and fluctuating base pairs [86]. On the challenging multi-objective optimization task [54] of designing molecules with desired properties, we demonstrate superior control over molecule properties during the generation in a decoder-only setting compared to a state-of-the-art vanilla Transformer architecture. We want to point out that molecular research inevitably bears the risk of misuse [87], but we strongly distance ourselves from any such attempts.

However, our approach also has limitations. The additional layer and the posterior model increase the computational and memory needs up to a factor of two. Our approach is also limited to encoder-only or decoder-only setups since it requires a target of the same length as the input sequence. We have not applied our model in natural language processing yet, e.g. as a language model, which could improve predictive performance and save compute at inference time due to the probabilistic encoder. Another future application could be in text-to-speech or automated speech recognition since natural speech contains many ambiguities. In general, our approach and especially the conditional variational training using the GECO objective and kappa annealing, could be used in future probabilistic models. Kappa annealing reduces the need for exhaustive hyperparameter optimization and increases performance. Therefore, we expect that the positive impact of our work is not limited to RNA folding and molecule design but also to generative models in general.

## Acknowledgments and Disclosure of Funding

Jörg Franke, Frederic Runge, and Frank Hutter acknowledge funding by European Research Council (ERC) Consolidator Grant "Deep Learning 2.0" (grant no. 101045765). Funded by the European Union. Views and opinions expressed are however those of the author(s) only and do not necessarily reflect those of the European Union or the ERC. Neither the European Union nor the ERC can be held responsible for them.

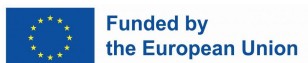

A part of this work was supported by the German Federal Ministry of Education and Research (BMBF, grant RenormalizedFlows 01IS19077C) and by the German Research Foundation (DFG, grant no. 417962828). Furthermore, the authors acknowledge support by the state of Baden-Württemberg through bwHPC and the DFG (grant no. INST 39/963-1 FUGG). We also thank Tom Kaminski, Fabio Ferreira, Mahmoud Safari and Arbër Zela for useful comments on the manuscript.

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
