# Appendix

## A  Training Dynamics of ProbTransformer

In this section, we provide insights into the training dynamics of the ProbTransformer using the RNA folding task. In Figure 5 we visualize the progress of the training. The first row pictures the hamming distance on the validation set, in the second we show the learning rate schedule, in the third the annealing of kappa, in the fourth the cross-entropy loss $\mathcal{L}_{rec}$, in the fifth the KL loss $\mathcal{D}_{KL}$, in the sixth the adjustment of $\kappa$, and at the bottom the reconstruction constraint $\mathcal{L}_{rec} - \kappa$.

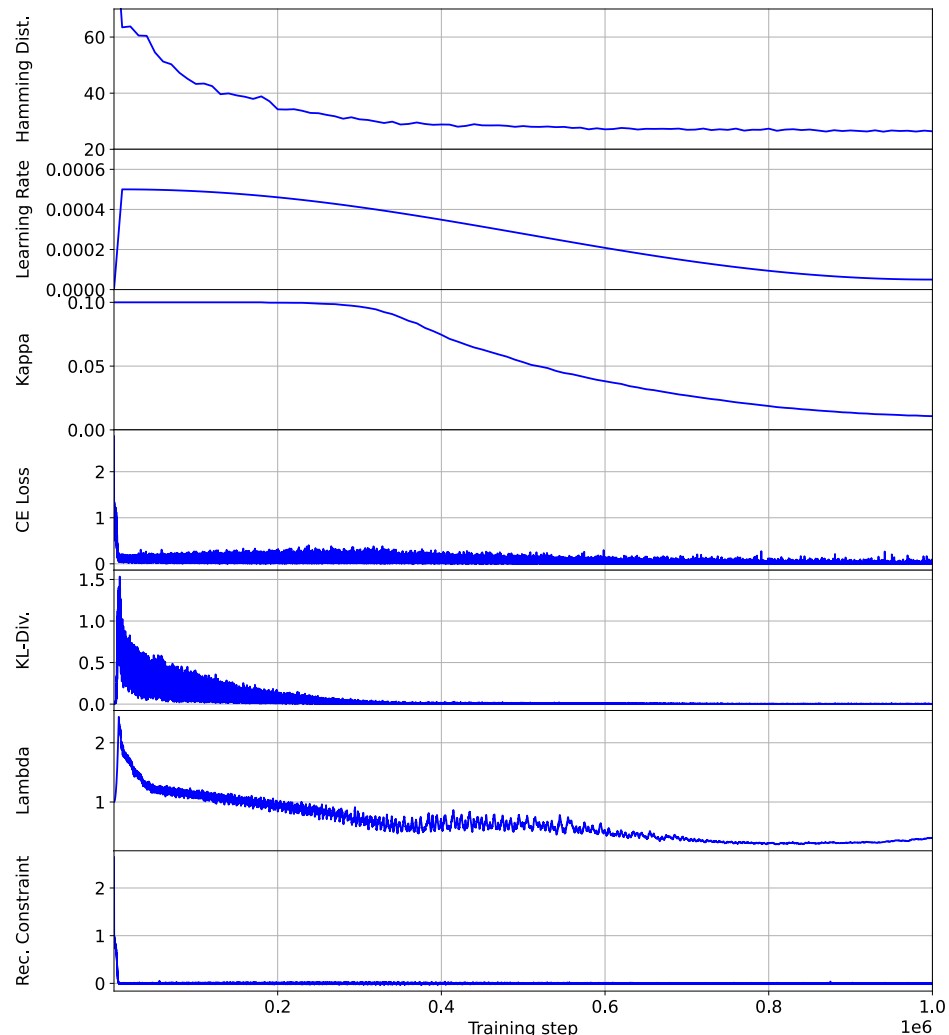

Figure 5: The training dynamics of the ProbTransformer in the RNA folding task.

We observe that the $\kappa$ decreases over time due to a low $\lambda$ which allows an increase in the pressure on the reconstruction. In Figure 6, we show the same training but with a log scale on the x-axis to focus on the early training phase. At the beginning of the training, the reconstruction constraint is not satisfied and the Lagrange multiplier $\lambda$ is increasing which results in pressure on the reconstruction loss. At the same time, the KL divergence increases due to reconstruction via $\mathbf{Z}^{post}$, leading to an increase in the initial distance of $P_\phi$ and $Q_\psi$. Once the reconstruction constraint is satisfied, $\lambda$ decreases and the pressure moves to the KL term. Also, the performance, measured in terms of the Hamming distance, does not improve even when the CE loss drops, since the CE loss only trains the posterior reconstruction. It only starts to improve when the KL divergence begins to decrease and the predictive model learns to create a useful internal latent representation.

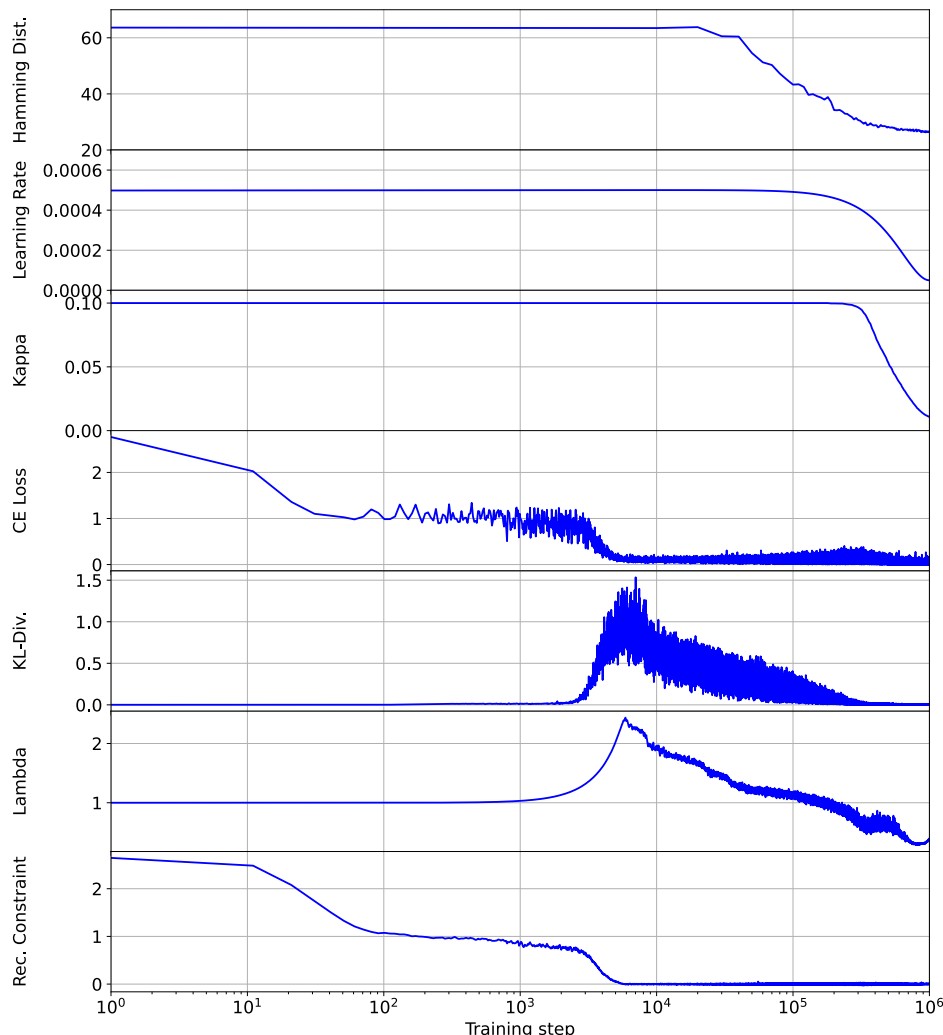

Figure 6: The training dynamics of the ProbTransformer in the RNA folding task but with log scale on the x axis to focus on the early phase of the training.

# B Synthetic Sequential Distribution Task

This section provides more details about the synthetic sequential distribution task itself, the configuration of the used Transformer and ProbTransformer model, the training process, and the results.

## B.1 Data

We design the task to map a sequence of tokens from a source vocabulary $x \in \mathcal{V}_i^*$ to a sequence of target tokens from a target vocabulary $y \in \mathcal{V}_o^*$ with the same length. The tokens in the source sequence are used to build 'phrases" $\mathbb{P}$. Each phrase consists of $l$ tokens sampled with replacement (similar to the combination of words in a sentence). We randomly generate a unique distribution $p(y|x, \mathbb{P})$ over the target tokens for each source token in each phrase, depending on the current phrase. Further, we design the distribution sparsely so that no more than $k$ tokens from the target vocabulary have a non-zero probability. The training data is generated by sampling input sequences from all phrases (with replacement) and sampling the target sequence from its corresponding distribution. The size of the source and target vocabulary is 500, a phrase exists of three tokens, and we create 1000 different sections. Each target token is drawn from a sparse distribution with 1 to 10 non-zero token probabilities. The sequence length is uniformly drawn from a length of 15 to 90. We created 100.000 training samples and 10.000 validation and test samples. Please find the detailed configuration of the task in Table 5.

Table 5: Configuration of the synthetic sequential distribution task.

| | |
|---|---:|
| Max token length | 90 |
| Min. token length | 15 |
| Number of phrases | 1000 |
| Number of training samples | 100000 |
| Token per phrase $l$ | 3 |
| Possible target tokens $k$ | 10 |
| Vocabulary source tokens | 500 |
| Vocabulary target tokens | 500 |

Figure 7 shows an example of target distribution depending on the source phrase. On the x-axis, we show the target vocabulary consisting of number-tokens. On the y-axis, there are two phrases of source tokens. The yellow-green-blue color scheme represents the distribution of the target token mapping to a source token depending on the source phrase. Please note that the tokens in the second and third rows are the same but have different distributions due to the position in the phrase. A target sequence is sampled from this distribution, and in the optimal case, the model should be able to reproduce this distribution.

## B.2 Setup

In general, we implement our models and tasks in Python 3.8 using mainly PyTroch [88], Numpy [89], and Pandas [90]. We use Matplotlib [91] for the plots in the paper.

For experiments on the synthetic sequential distribution task, we use the configuration listed in Table 6 for the Transformer and ProbTransformer. For MC dropout, we employed a grid search to find the optimal Dropout rate $(0.1, 0.2, \cdots, \mathbf{0.5})$. The other hyperparameters were tuned manually based on preliminary work [1] or based on preliminary experiments. We use SiLU [92] as activation function in both models. Furthermore, we use automatic mixed-precision during the training, initialize the last linear of each layer (feed-forward, attention, or prob layer) with zero, and use a learning rate warm-up in the first epoch of training as well as a cosine learning rate schedule. We use the squared softplus function to ensure a positive $\lambda$ value during training and update $\lambda$ with a negative gradient scaling of $0.01$. For the moving average of the reconstruction loss, we use a decay of $0.95$.

## B.3 Results

We provide detailed results of our models and sampling methods with mean and standard deviation for five random seeds and evaluate two additional metrics: (1) We count the different output variations.

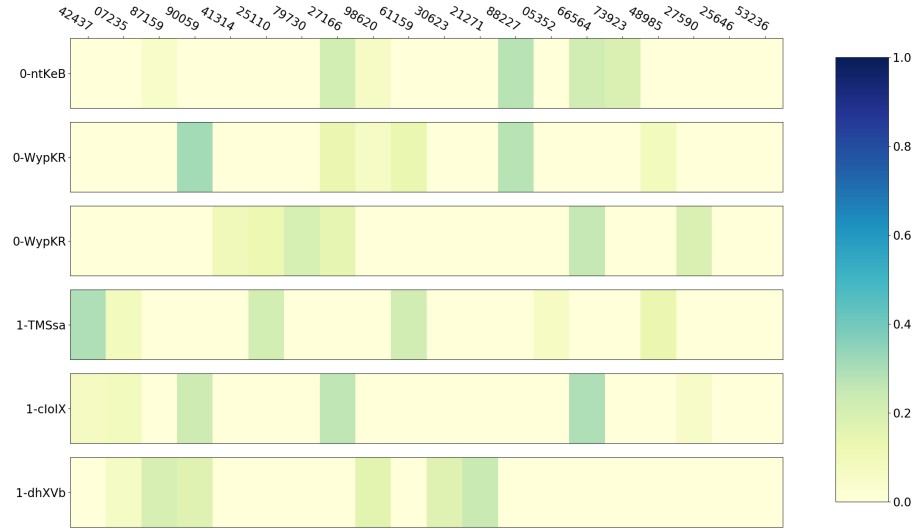

Figure 7: Target distribution depending on the source phrase.

Table 6: Hyperparameters of the Transformer and ProbTransformer training in the synthetic sequential distribution task.

| | |
|---|---|
| Feed-forward dim | 1024 |
| Latent Z dim | 256 |
| Model dim | 256 |
| Number of layers | 4 |
| Number of heads | 4 |
| Prob layer | all layer |
| Kappa | 0.1 |
| Dropout | 0.1 |
| Optimizer | adamW |
| Beta 1 | 0.9 |
| Beta 2 | 0.98 |
| Gradient Clipping | 100 |
| Learning rate schedule | cosine |
| Learning rate high | 0.001 |
| Learning rate low | 0.0001 |
| Warmup epochs | 1 |
| Weight decay | 0.01 |
| Epochs | 50 |
| Training steps per epoch | 2000 |

A perfect model creates the same *diversity* as nonzero probabilities in the true distribution. We normalize this measure to one; high values suggest more different tokens than non-zero tokens in the true distribution, and smaller values suggest fewer tokens. (2) Another measure for the distance between two distributions is the *total variation* which can deal with zero probabilities. Please find the results in Table 7.

Table 7: The mean and standard deviation of five random seeds for runs with Transformer and ProbTransformer in the Synthetic Sequential Distribution Task.

| Model | | Validity | | Diversity | | KL-divergence | | Total Variation | |
|---|---|---|---|---|---|---|---|---|---|
| | | mean | std | mean | std | mean | std | mean | std |
| ProbTransformer | | 0.99 | 0.0024 | 0.99 | 0.0002 | 0.52 | 0.0165 | 0.11 | 0.0007 |
| Transformer | dropout | 0.93 | 0.0198 | 0.72 | 0.0014 | 12.71 | 0.0357 | 0.35 | 0.0004 |
| | softmax | 0.73 | 0.0075 | 0.90 | 0.0020 | 7.84 | 0.0654 | 0.31 | 0.0016 |

# C  RNA Folding

In this section, we detail our data pipeline, the general experimental setup and evaluation protocol, and show additional results, including standard deviations for multiseed runs, for our experiments on the RNA folding problem. We start, however, with a brief introduction to RNA functions and the importance of their secondary structure.

RNAs are one of the major regulators in the cell and have recently been connected to diseases like cancer [93] or Parkinson's [94, 95]. They consequently arise as a promising alternative for the development of novel drugs, including antiviral therapies against COVID-19[96] and HIV[97], or vaccines[98].

The vast majority of RNAs that are differentially transcribed from the human genome do not encode proteins [99, 100] and revealing the functions of these so-called non-coding RNAs (ncRNAs) is one of the main challenges for understanding cellular regulatory processes [31]. Similar to proteins, the function of an RNA molecule strongly depends on its folding into complex shapes, but unlike protein folding, which is dominated by hydrophobic forces acting globally, RNAs exhibit a hierarchical folding process [101]. In a first step, the corresponding nucleotides of the RNA sequence connect to each other by forming hydrogen bonds, resulting in local geometries and a distinct pairing scheme of the so-called secondary structure of RNA[3]. The secondary structure defines the accessibility of regions for interactions with other cellular compounds [31] and dictates the formation of the 3-dimensional tertiary structure [101, 102]. However, RNA structures are highly dynamic, which dramatically influences their functions [13, 14]. A learning algorithm that tackles the problem of predicting these structure ensembles is currently lacking in the field and we consider our work a major step in the direction of accurate RNA structure prediction.

## C.1  Data

In this section, we detail the datasets used during training and for our experiments. RNA sequences are chains of the four nucleotides (bases) *adenine*, *cytosine*, *guanine*, and *uracil*. However, RNA data often considers an extended nucleotide alphabet using IUPAC nomenclature[4] and we note that the datasets used in this work include IUPAC nucleotides.

A RNA secondary structure is typically described as a list of pairs where a pair $(i, j)$ denotes two nucleotides at the positions $i$ and $j$ of a RNA sequence that are connected by hydrogen bonds to form a base pair. In the simplest case, all pairs of the secondary structure are nested, i.e. if $(i, j)$ and $(k, l)$ describe two pairs of a secondary structure with $i < k$, then $i < k < l < j$. A functional important class of base pairs [47, 48], however, is called pseudoknots, where the nested pairing

---

[3]We note that there is a longstanding discussion in the field of structural biology on what is called an RNA secondary structure and we use the broadest definition of secondary structure, i.e. including non-nested structures, in this work.

[4]We refer to the IUPAC nomenclature described by the International Nucleotide Sequence Database Collaboration (INSDC) at `https://www.insdc.org/documents/feature_table.html#7.4.1`.

scheme is disrupted by one or more pairs of type: $i < k < j < l$. Canonical base pairs are formed between A and U, G and C (Watson-Crick pairs) or between G and U (Wobble pairs), while all other pairings of nucleotides are called non-canonical base pairs. We use the dot-bracket notation [12] for description of secondary structures where a dot corresponds to unpaired nucleotides and a pair of matching brackets denotes a pair of two nucleotides.

For our experiments we collect a large pool of annotated RNA secondary structures and their corresponding sequences from recent publications [41, 39, 37, 103, 38]. In particular, we collect 102098 samples from the BpRNA [103] meta database, two versions of the RNAStralign [104] dataset provided by [41] and [39] with 28168 and 20897 samples, respectively, two versions of the ArchiveII [105] dataset provided by [41] and [39] with 2936 and 3966 samples, the TR0, VL0, and TS0 datasets provided by [37] with 10814, 1300, and 1305 samples, respectively, the TrainSetA [106] and the TrainSetB [106] with 3164 and 1094 samples, respectively, and all available data from the RNA-Strand [38] database (3898 samples). For all data provided in `.bpseq`, .ct or similar file formats that only provide base pairs, we use BpRNA [103] to consistently annotate secondary structures with our major data source, the BpRNA metadatabase. We split the testset TS0 and the validation set VL0 from the pool and uniformly sampled a novel testset, TSsameSeq, from sequences of the remaining pool as described in Section 4.2. The highly redundant raw data consists of 177035 training samples, 1300 validation samples and 1351 test samples. We remove duplicates from the data as well as samples that did not contain any pairs. We applied `CD-HIT-EST-2D` [107] to remove sequences from the training data with a sequence similarity greater than 80% to the validation and test samples, the lowest available threshold [37]. In accordance to [37], we limit the length of sequences to 500 nucleotides to save computational budget and since especially for longer RNAs, experimental evidence is generally still lacking because of challenges in crystallization and spectral overlap [108]. Table 8 summarizes the final datasets used for our experiments.

Table 8: Statistics of the different datasets used for our experiments on RNA folding.

| Dataset | # Samples | Unique Seq. | Unique Struc. | Avg. Length | Pair Types | | |
|---|---|---|---|---|---|---|---|
| | | | | | Canonical | Non-Canonical | Pseudoknots |
| Train | 52007 | 48092 | 27179 | 137.46 | 1701469 | 106208 | 47382 |
| VL0 | 1299 | 1299 | 1218 | 131.94 | 35301 | 4096 | 1001 |
| TS0 | 1304 | 1304 | 1204 | 136.09 | 36702 | 4083 | 1206 |
| TSsameStruc | 149 | 149 | 49 | 85.04 | 2849 | 211 | – |
| TSsameSeq | 46 | 20 | 46 | 176.46 | 2273 | 60 | 150 |

## C.2    Setup

In this section, we provide the configuration of the models and training details. We list the hyperparameters of the Transformer and ProbTransformer training in the RNA folding experiment in Table 9. We use the squared softplus function to ensure a positive $\lambda$ value during training and update $\lambda$ with a negative gradient scaling of $0.1$. For the moving average of the reconstruction loss we use a decay of $0.95$. We performed the training on one Nvidia RTX2080 GPU and the training time for one ProbTransformer model is $\sim$63h and for one Transformer $\sim$33h. The training time for the ProbTransformer nearly doubles due to the posterior model.

### C.2.1    CNN Head

Although the Transformer's and ProbTransformer's output prediction has a high quality, its still sometimes flawed. This hinders the evaluation of the F1 score and therefore the comparison on this metric to related work. Instead of manually designing an error correction heuristic we decided to learn a simple model which takes the Transformer's last latent and predicts an adjacency matrix which is use to evaluate the F1 score of our prediction.

We use a fixed-size CNN without up or down scaling. The detailed hyperparameters and training configuration of our CNN is listed in Table 10. The input is a concatenation of the vertical and horizontal broadcast of the last latent from the Transformer as well as the embedded nucleotide sequence. The output are two classes, one for a connection between nucleotides and one for no connection. We train our CNN head on the same training data as the Transformer and pre-compute the Transformer output to save computational resources during the CNN training, i.e. we do not train

Table 9: Hyperparameters of the Transformer and ProbTransformer and training details of the RNA folding experiment.

| | |
|---|---|
| Feed-forward dim | 2048 |
| Latent Z dim | 512 |
| Model dim | 512 |
| Number of layers | 6 |
| Number of heads | 8 |
| Prob layer | 2,3,4,5 |
| Kappa | 0.1 |
| Dropout | 0.1 |
| Optimizer | adamW |
| Beta 1 | 0.9 |
| Beta 2 | 0.98 |
| Gradient Clipping | 100 |
| Learning rate schedule | cosine |
| Learning rate high | 0.0005 |
| Learning rate low | 0.00005 |
| Warmup epochs | 1 |
| Weight decay | 0.01 |
| Epochs | 100 |
| Training steps per epoch | 10000 |

them jointly. We use early stopping based on the Hamming distance of the validation set. We perform the training on one Nvidia RTX2080 GPU and the training time is ∼3h.

Table 10: Hyperparameters of the CNN head and training configuration.

| | |
|---|---|
| Model dim | 64 |
| Number of layers | 8 |
| Stride | 1 |
| Kernel | 5 |
| Dropout | 0.1 |
| Optimizer | adamW |
| Beta 1 | 0.9 |
| Beta 2 | 0.98 |
| Gradient Clipping | None |
| Learning rate schedule | cosine |
| Learning rate high | 0.005 |
| Learning rate low | 0.0005 |
| Warmup epochs | 1 |
| Weight decay | 1e-10 |
| Epochs | 10 |
| Training steps per epoch | 2000 |

## C.3   Results

In this section, we describe our evaluation protocol in detail and show additional results for the predictions on the three test sets, TS0, TSsameStruc, and TSsameSeq, including standard deviations. For drawing of RNA secondary structures, we use `VARNA` [109] provided under GNU GPL License.

## C.3.1   Evaluation

We evaluate all approaches concerning Hamming distance, the number of solved tasks, and the F1 score. The Hamming distance is the raw count of mismatching characters in two strings of the same length. A dot-bracket structure with a Hamming distance of zero counts as solved. F1 score describes

the harmonic mean of precision (PR) and sensitivity (SN) and is computed as follows:

$$PR = \frac{TP}{(TP + FP)} \quad , \tag{12}$$

$$SN = \frac{TP}{(TP + FN)} \quad , \tag{13}$$

$$F1 = 2 \cdot \frac{(PR \cdot SN)}{(PR + SN)} \quad , \tag{14}$$

where TP, FP and FN denote true positives, false positives and false negatives, respectively. For TSsameSeq, we only evaluate the best predictions concerning Hamming distance.

**SPOT-RNA**  The output of SPOT-RNA is in `.ct` tabular format with columns for indication of pairs. Deriving pseudoknots from base pairs is not trivial [110, 103] and we, therefore, convert the output to `.bpseq` format and apply BpRNA [103], the same annotation tool as we used during data generation, to yield annotated secondary structures for all predictions of SPOT-RNA.

**MXFold2**  MXFold2 directly outputs secondary structures in dot-bracket format, which we evaluate directly.

**RNAfold**  As for MXFold2, RNAfold's predictions can be evaluated directly from the output in dot-bracket format.

We note that RNAfold and MXFold2 are not capable of predicting pseudoknots due to their underlying dynamic programming approach.

**UFold**  In contrast to all other approaches, UFold cannot handle IUPAC nucleotides in the input sequences. When evaluating the exact same test data used for all other approaches, the recommended webserver of UFold at `https://ufold.ics.uci.edu/` as well as the standalone version generates predictions with different lengths compared to the inputs which cannot be evaluated. A fair comparison with UFold thus was not possible and we decided to exclude UFold from the evaluations in the main paper. However, we resolved IUPAC nucleotides by uniformly sampling corresponding canonical nucleotides for IUPAC nucleotides to create a dataset accepted by UFold. We use the dot-bracket output of UFold for evaluations since provided `.ct` files resulted in errors when trying to obtain secondary structures using BpRNA similar as described for SPOT-RNA due to predictions with nucleotides pairing with themselves. The results of UFold on TS0 and TSsameStruc are shown in Table 11.

Table 11: Structure fidelity of UFold and the ProbTransformer on TS0 and TSsameStruc.

| Model | TS0 | | | TSsameStruc | | |
|---|---|---|---|---|---|---|
| | F1 Score | Hamming | Solved | F1 Score | Hamming | Solved |
| ProbTransformer | **62.5** | **27.4** | **0.118** | **93.2** | **3.2** | **0.550** |
| UFold | 58.8 | 33.3 | 0.038 | 82.7 | 9.4 | 0.141 |

**ProbTransformer**  Regarding Hamming distance and the number of solved structures, we directly evaluate the predictions of the ProbTransformer from the raw model outputs. For the F1 Score, however, we use a post-processing step using the CNN head to obtain an adjacency matrix as described above.

**Structure Ensemble Predictions**  We use the dot-bracket output of all approaches for evaluations on TSsameSeq. The predictions with the lowest Hamming distance to the respective ground truth structure were used for the evaluation of performance.

### C.3.2  Detailed results

In this section, we provide further results for our experiments on the RNA folding problem, including standard deviations for multiseed runs.

**TS0** We provide additional results for predictions on TS0. In Table 12 we provide results with standard deviations. Figure 8, 9 and 10 show example predictions of different approaches. Figure 13 shows the F1 score for different base-pairs in comparison with related work.

Table 12: Mean and standard deviation for three random seeds of the ProbTransformer and Transformer on TS0.

| Model | TS0 | | | | | |
|---|---|---|---|---|---|---|
| | F1 Score | | Hamming | | Solved | |
| | mean | std | mean | std | mean | std |
| ProbTransformer | 62.5 | 0.004 | 27.4 | 0.3425 | 0.118 | 0.0037 |
| Transformer | 50.5 | 0.0109 | 35.27 | 0.5911 | 0.084 | 0.0019 |

Table 13: The F1 score of different base-pairs of the ProbTransformer, Transformer and related work on TS0.

| Method | F1-All | F1-WC | F1-canonical | F1-wobble | F1-NC |
|---|---|---|---|---|---|
| ProbTransformer | 62.5 | 65.7 | 64.7 | 58.1 | 39.5 |
| Transformer | 50.5 | 53.4 | 52.5 | 47.7 | 36.3 |
| SPOT-RNA | 59.7 | 63.7 | 62.7 | 43.9 | 15.6 |
| MXFold2 | 55.0 | 58.0 | 57.0 | 41.7 | 0.0 |
| RNAFold | 49.2 | 52.0 | 50.8 | 36.9 | 0.0 |

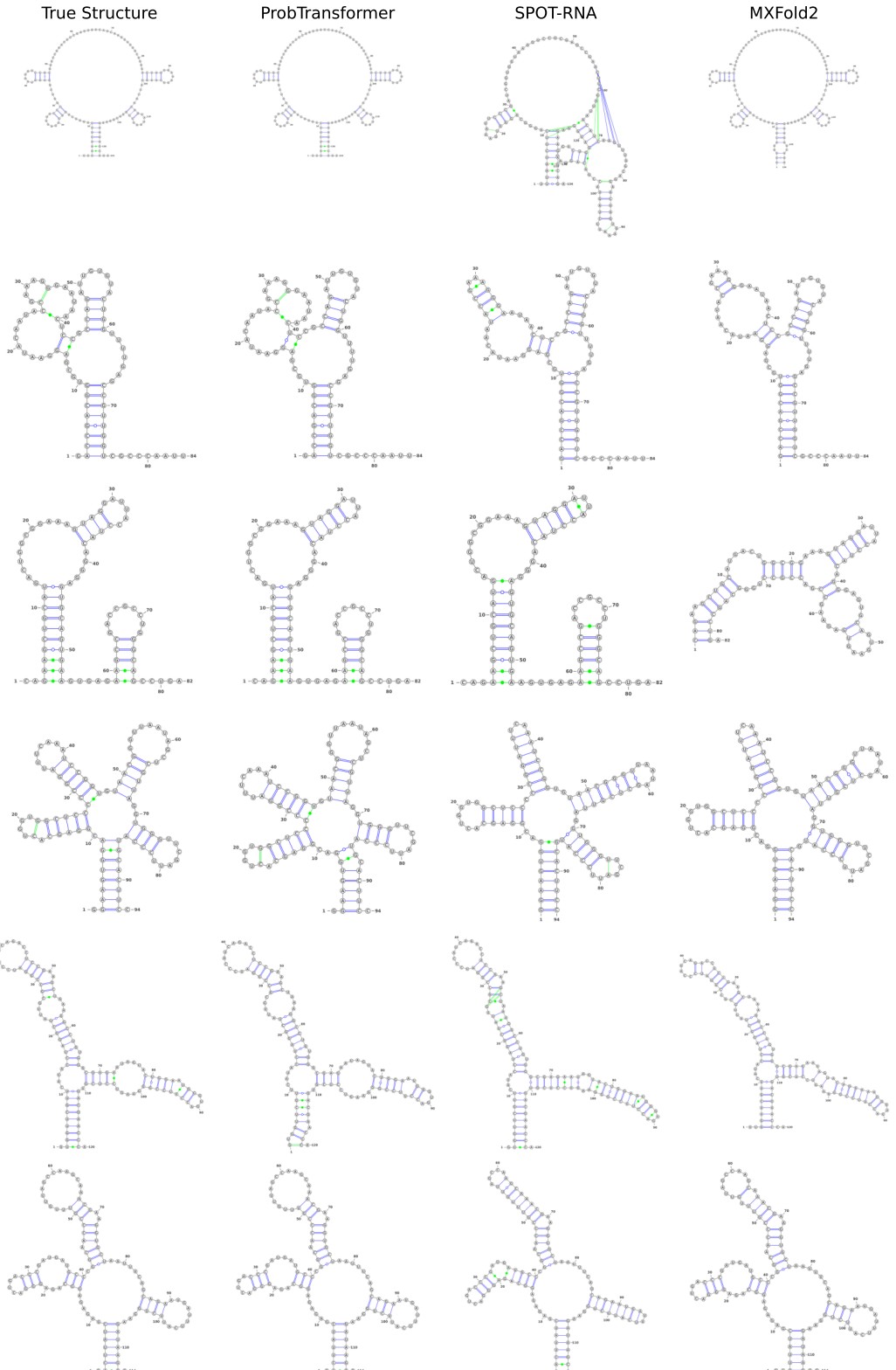

Figure 8: RNA Structure prediction examples for the test set TS0. The shown structures for the ProbTransformer are derived from the raw model outputs without further post-processing.

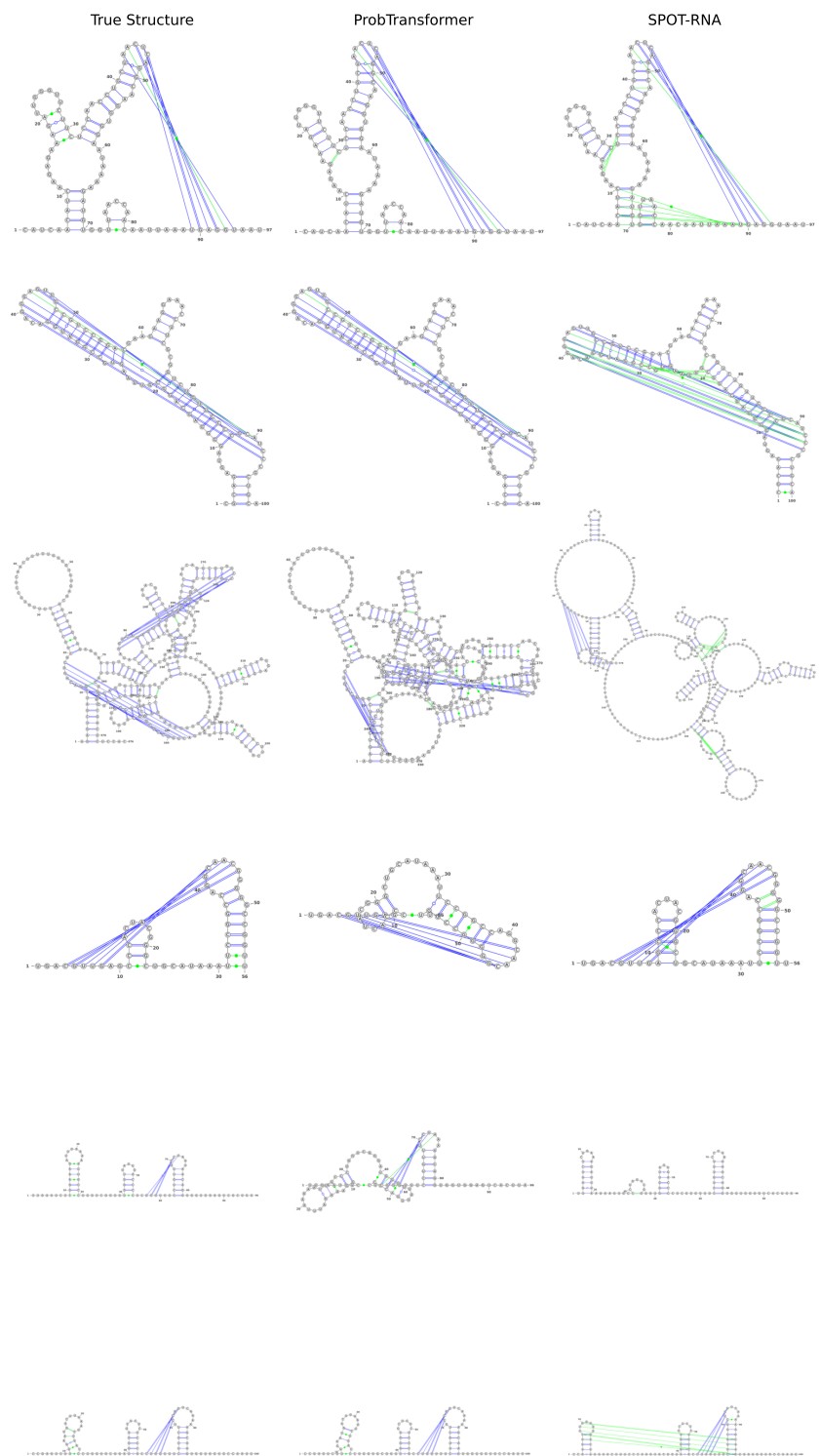

Figure 9: RNA Structure prediction examples for targets that contain pseudoknots from the test set TS0. The shown structures for the ProbTransformer are derived from the raw model outputs without further post-processing. We only show the two algorithms that are capable of predicting pseudoknots.

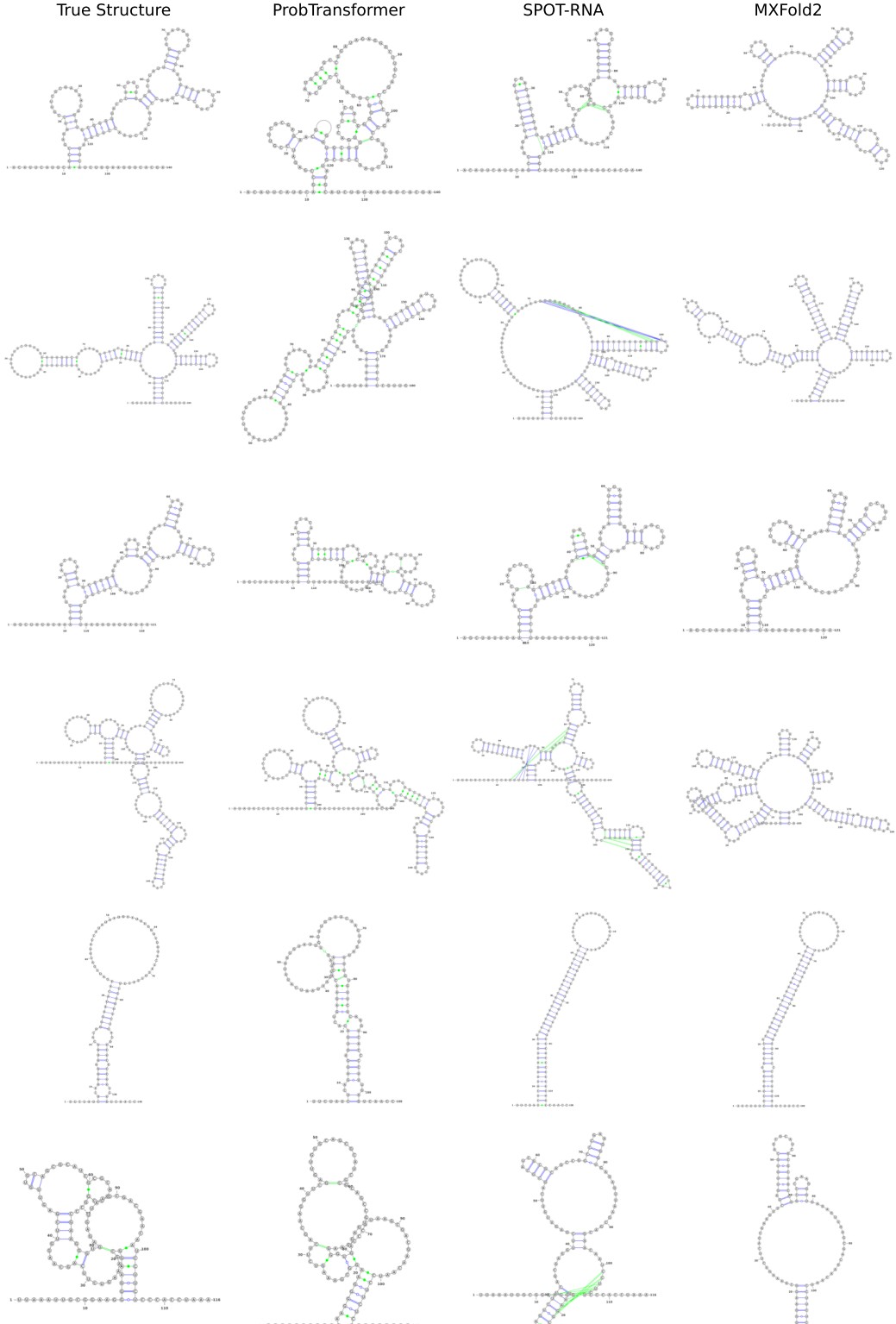

Figure 10: RNA Structure prediction examples for the test set TS0 for inaccurately predicted structures. The shown structures for the ProbTransformer are derived from the raw model outputs without further post-processing.

**TSsameStruc** We provide additional results for predictions on TSsameStruc. Table 14 shows results with standard deviation.

Table 14: Mean and standard deviation for three random seeds of the ProbTransformer and Transformer on TSsameStruc.

| Model | TSsameStruc | | | | | |
|---|---|---|---|---|---|---|
| | F1 Score | | Hamming | | Solved | |
| | mean | std | mean | std | mean | std |
| ProbTransformer | 93.2 | 0.005 | 3.22 | 0.163 | 0.55 | 0.0055 |
| Transformer | 89.5 | 0.0111 | 4.55 | 0.0316 | 0.481 | 0.0084 |

**TSsameSeq** We provide results for the predictions on TSsameSeq to analyze the ability of the ProbTransformer to capture the structure distribution of RNA sequences that map to different structures. For all experiments, we inferred the model 5, 10, 20, 50, and 100 times using sample inference and analyzed the raw predictions without further post-processing. We observe that the ProbTransformer has learned the structure distributions from the data, producing predictions closer to the desired structures as indicated by a low Hamming distance shown in Table 15. The mean and standard deviations of the predictions are shown in Table 17, results for individual samples are summarized in Table 16. Remarkably, the ProbTransformer is the only model that reproduces both true structures for two of the 20 RNA sequences from the raw model predictions directly using only 5 inferences (with two out of three random seeds, results not shown).

Table 15: Average minimum Hamming distance of the different approaches on TSsameSeq.

| Model | Hamming Distance | | | | |
|---|---|---|---|---|---|
| | N=5 | N=10 | N=20 | N=50 | N=100 |
| ProbTransformer | **26.51** | **25.16** | **24.47** | **23.60** | **23.09** |
| Transformer | 49.17 | 49.17 | 49.17 | 49.17 | 49.17 |
| RNAsubopt | 42.59 | 42.83 | 38.09 | 34.22 | 31.30 |
| RNAshapes | 47.65 | 45.83 | 45.24 | 39.04 | 37.59 |
| RNAstructure | 47.22 | 42.02 | 38.04 | 35.20 | 32.59 |

Table 16: Minimal Hamming distances per Structure for all samples of TSsameSeq. We show results for one random seed of the ProbTransformer only.

| Family | #Structures | ProbTransformer | RNAsubopt | RNAshapes | RNAStructure |
|---|---|---|---|---|---|
| 5S rRNA | 2 | 0/0 | 8/12 | 12/16 | 8/12 |
| 5S rRNA | 2 | 8/0 | 16/12 | 14/10 | 24/22 |
| 5S rRNA | 2 | 8/0 | 10/2 | 12/4 | 12/4 |
| Group I catalytic intron | 2 | 24/24 | 44/47 | 51/53 | 50/56 |
| N/A | 2 | 76/88 | 51/65 | 59/71 | 60/72 |
| Antizyme RNA frameshifting stimulation element | 2 | 14/2 | 12/0 | 16/6 | 15/4 |
| N/A | 3 | 7/5/1 | 6/4/0 | 6/4/0 | 6/4/0 |
| tRNA | 2 | 0/4 | 4/0 | 4/0 | 4/0 |
| transfer-messenger RNA | 3 | 17/33/29 | 91/62/112 | 140/111/152 | 107/77/119 |
| tRNA | 2 | 0/2 | 2/0 | 2/0 | 2/0 |
| N/A | 2 | 14/13 | 12/2 | 12/2 | 10/0 |
| Bacterial small signal recognition particle RNA | 2 | 6/7 | 10/6 | 10/6 | 10/6 |
| Hammerhead ribozyme (type I) | 2 | 36/35 | 14/14 | 8/8 | 8/8 |
| Group I catalytic intron | 2 | 118/116 | 54/53 | 102/101 | 66/65 |
| 5S rRNA | 2 | 9/1 | 10/6 | 10/6 | 12/8 |
| 5S rRNA | 2 | 11/1 | 12/12 | 16/8 | 20/14 |
| Bacterial RNase P class A | 3 | 14/15/15 | 63/60/52 | 55/51/46 | 61/59/54 |
| Bacterial RNase P class A | 3 | 9/6/6 | 50/47/42 | 61/58/55 | 43/43/38 |
| Bacterial RNase P class B | 4 | 65/66/68/69 | 78/79/101/99 | 80/78/104/101 | 65/67/89/91 |
| 5S rRNA | 2 | 3/1 | 4/0 | 6/2 | 4/0 |

Table 17: Mean and standard deviation of the ProbTransformer and Transformer on TSsameSeq for three random seeds.

| Model | TSsameSeq | | | | | | | | | |
|---|---|---|---|---|---|---|---|---|---|---|
| | N=5 | | N=10 | | N=20 | | N=50 | | N=100 | |
| | mean | std | mean | std | mean | std | mean | std | mean | std |
| ProbTransformer | 26.51 | 0.7754 | 25.16 | 0.4687 | 24.47 | 0.3388 | 23.60 | 0.2310 | 23.09 | 0.4820 |
| Transformer | 49.17 | 2.6304 | 49.17 | 2.6304 | 49.17 | 2.6304 | 49.17 | 2.6304 | 49.17 | 2.6304 |

## C.4 Related work RNA folding

In this section, we discuss state-of-the-art deep learning approaches for the RNA folding problem in detail.

*SPOT-RNA* [37] was the first algorithm using deep neural networks for end-to-end prediction of RNA secondary structures. In this work, an ensemble of residual networks (ResNets) [111] and bidirectional LSTMs [112] (BiLSTMs) [113] was pre-trained on a large set of RNA secondary structure data and then fine-tuned on a small set of experimentally-derived RNA data, including tertiary interactions. Although the authors claimed the possibility of predicting RNA tertiary interactions, the performance for these types of base pairs was poor and the currently available version of the algorithm excludes tertiary interactions from its outputs. We thus consider this work as RNA secondary structure prediction.

*E2efold* [41] uses a Transformer encoder architecture to learn the prediction of RNA secondary structures. The algorithm was trained on a very homologous set of RNA data and showed strongly reduced performance when evaluated on data of other publications [39, 42], indicating strong overfitting. Since we use the same data set as the respective work, we exclude *E2efold* from our evaluations.

*MXFold2* [39] combines deep learning with a DP approach by using a CNN/BiLSTM architecture to learn the scoring function for the DP algorithm. The network is trained to predict scores close to a set

of thermodynamic parameters to increase robustness. *MXFold2* is restricted to predict a reduced set of base pairs due to limitations in the DP algorithm.

*UFold* [42] employs a UNet [114] architecture for solving the RNA folding problem. Similar to *SPOT-RNA*, the authors additionally report results for predictions on data that contains tertiary interactions after fine-tuning the model on experimental data with slightly worse overall performance compared to *SPOT-RNA*. In contrast to the previously described works, however, *UFold* treats an RNA sequence as an image of all possible base-pairing maps (16 maps corresponding to 16 possible pairs) and an additional map for pair probabilities, represented as square matrices of the provided sequence.

## D   Molecular Design

In this section, we provide further information on our experiments for the conditional generation of molecules based on multiple desired properties.

Estimations of the size of the chemical space [52] vary widely [115] (typically between $10^{20}$ and $10^{100}$) with a common consensus that it contains too many molecules to be explicitly enumerated [53]. Deep generative models recently attracted huge interest in exploring this practically infinite space for the use in drug discovery and deep learning-based molecular *de novo* generation has emerged as the most interesting and fast-moving field in cheminformatics [53] during the last five years. In this so-called generative chemistry [51], deep generative models are typically trained on a large part of enumerated chemical space to learn a biased distribution of molecular representations and evaluated for their ability to generate novel compounds and explore the unseen chemical space. Common evaluation protocols include metrics to measure e.g. the *novelty* of the designed compounds concerning the examples visited during training, their *uniqueness* to measure the internal diversity of predictions, and *validity* of the generated compounds regarding e.g. the underlying SMILES grammar [57]. However, besides general exploration which could be achieved using uniform sampling approaches [53], biological applications typically require that the designed molecules have certain desired properties. For generative models, the task is then to explore the chemical space conditioned on molecule properties (conditional generation).

### D.1   Data

For sequence-based approaches, a common way of representing molecules is the simple molecular line-entry system (SMILES) [15]. This notation was originally proposed to represent molecules as strings and uses a sequence of elements combined with special characters to enable branching, ring-closure, and different bond orders as well as indications for properties like charges [53]. To train the ProbTransformer, we use the training data of the GuacaMol [57] benchmark suite provided by [11]. Overall, the training data consists of 1259543 SMILES with a vocabulary of 94 unique characters.

### D.2   Setup

In this section, we provide the configuration of the models and training details. We list the hyperparameters of the Transformer and ProbTransformer training for the molecule design experiment in Table 18. Furthermore, we use automatic mixed-precision during the training, initialize the last linear of each layer (feed-forward, attention, or prob layer) with zero, and use a learning rate warm-up in the first epoch of training. We use the squared softplus function to ensure a positive $\lambda$ value during training and update $\lambda$ with a negative gradient scaling of $0.01$. For the moving average of the reconstruction loss, we use a decay of $0.95$. We performed the training on one Nvidia RTX2080 GPU and the training time for one ProbTransformer model is $\sim$25h and for one Transformer $\sim$13h.

We condition the generation of molecules on three properties:

The *synthetic accessibility score (SAS)* is a measure of how difficult it is to synthesize a compound. The values for *SAS* could generally range between 1 (easy to synthesize) and 10 (very difficult to make).

The *partition coefficient (logP)* describes the logarithm of the partition coefficient of a compound. This measure compares the solubilities of a solute in two immiscible solvents at equilibrium. If one of the solvents is water and the other one is non-polar, *logP* is a measure of hydrophobicity.

Table 18: Hyperparameters of the Transformer and ProbTransformer and training details of the molecule design experiment.

| | |
|---|---:|
| Feed-forward dim | 1024 |
| Latent Z dim | 64 |
| Model dim | 256 |
| Number of layers | 8 |
| Number of heads | 8 |
| Prob layer | 2-7 |
| Kappa | 0.1 |
| Dropout | 0.1 |
| Optimizer | adamW |
| Beta 1 | 0.9 |
| Beta 2 | 0.98 |
| Gradient Clipping | 100 |
| Learning rate schedule | cosine |
| Learning rate high | 0.0005 |
| Learning rate low | 0.00005 |
| Warmup epochs | 1 |
| Weight decay | 0.01 |
| Epochs | 60 |
| Training steps per epoch | 5000 |

The *topological polar surface area (TPSA)* measures the ability of a drug to permeate cell membranes and describes the contributions of all polar atoms, such as oxygen and nitrogen and their attached hydrogens, to the molecular surface area. The polar surface area is a good estimator of the absorption, distribution, metabolism, excretion, and toxicology (ADMET)-relations of a compound and provides a rule-of-thumb for chemists to avoid dead-ends during the development process in drug discovery pipelines [116].

For our experiments, we follow the protocol described for *MolGPT* by choosing a value for each property from the following domains of values. *SAS*: 2.0, 4.0; *logP* 2.0, 6.0; *TPSA*: 40, 80. The model then conditionally generates molecules with the task to match all chosen values. Following *MolGPT*, results are reported in terms of the mean average deviation (MAD) and the standard deviation (SD) relative to the range of the desired property values. While our experiment focuses on conditional generation to match the desired property values, we also report the following scores.

*Validity* describes the fraction of generated molecules that are expressed as valid SMILES. High validity indicates strong learning of the underlying SMILES grammar.

*Uniqueness* is a measure of the prediction diversity. Uniqueness is the fraction of unique predictions from all generated valid SMILES. Although this metric is known to be not well-defined [16] and can be tricked by very simple means [58], we report uniqueness scores for reasons of comparability to previous work in the field.

*Novelty* is the fraction of valid molecules that are different from the training samples. A high novelty indicates strong exploration while a low novelty indicates overfitting.

We use `rdkit` [117] for computations of TPSA and logP and use the provided script by [11] for computations of SAS.

## D.3 Results

We provide detailed results with standard derivation in Table 19. Figure 11 shows the distributions of properties for all valid predictions of the ProbTransformer and MolGPT. We observe less deviation from the desired property values for the predictions of the ProbTransformer compared to MolGPT. In line with these results, we observe that the ProbTransformer generates more unique molecules with properties close to the desired property values compared to MolGPT as indicated in Table 20. Example predictions of these molecules are shown in Figures 12, 13, 14, 15, 16, 17, 18, and 19. We use `rdkit` [117] for drawing of the molecules.

Table 19: Results of the ProbTransformer in multi-property (TPSA+logP+SAS) conditional training on GuacaMol dataset on five different seeds.

| | Validity | Unique | Novelty | TPSA | | logP | | SAS | |
| | | | | MAD | SD | MAD | SD | MAD | SD |
|---|---|---|---|---|---|---|---|---|---|
| mean | 0.981 | 0.821 | 1 | 2.47 | 2.04 | 0.22 | 0.18 | 0.16 | 0.14 |
| std | 0.0123 | 0.0858 | 0 | 0.3727 | 0.3715 | 0.0544 | 0.0449 | 0.0767 | 0.0623 |

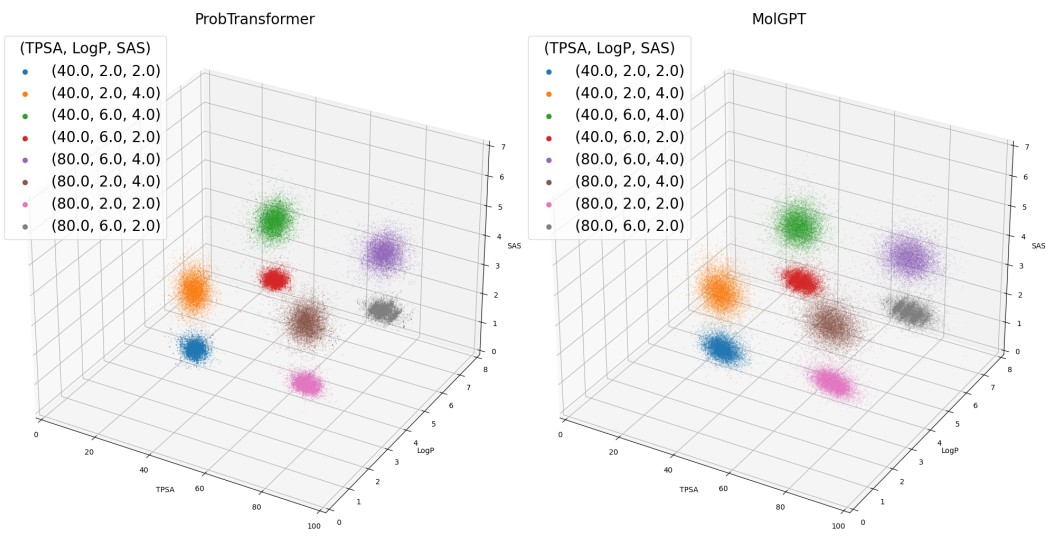

Figure 11: Distributions of predicted properties of the ProbTransformer and MolGPT. Results are shown for a representative seed for the ProbTransformer.

Table 20: Number of unique molecules that meet the desired properties for the ProbTransformer and MolGPT. We allow a deviation from the desired values of 0.5 for TPSA and 0.1 for SAS and LogP. Results show the mean with standard deviation for five random seeds of the ProbTransformer.

| (TPSA, LogP, SAS) | MolGPT | ProbTransformer | |
| | | Mean | Std |
|---|---|---|---|
| (40.0, 2.0, 2.0) | 58.0 | **80.4** | 4.1280 |
| (40.0, 2.0, 4.0) | 25.0 | **48.6** | 4.8415 |
| (40.0, 6.0, 4.0) | 20.0 | **40.4** | 3.7202 |
| (40.0, 6.0, 2.0) | 44.0 | **102.4** | 9.7693 |
| (80.0, 2.0, 2.0) | 105.0 | **220.2** | 16.8333 |
| (80.0, 2.0, 4.0) | 49.0 | **55.4** | 4.4091 |
| (80.0, 6.0, 4.0) | 50.0 | **60.8** | 6.2418 |
| (80.0, 6.0, 2.0) | 246.0 | **430.8** | 57.9876 |

TPSA: 40.0, LogP: 2.0, SAS: 2.0

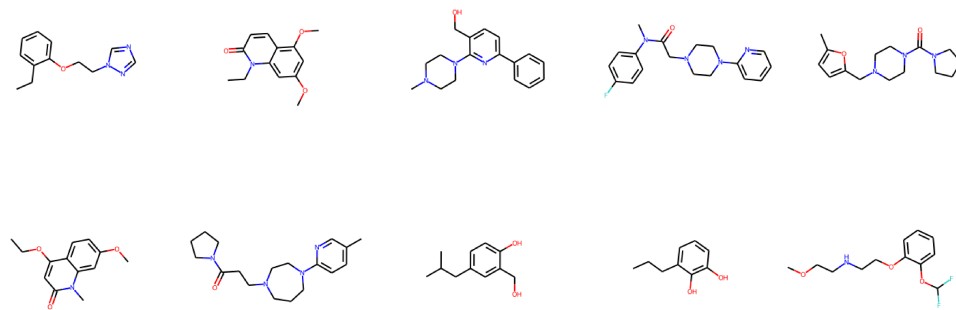

Figure 12: Example predictions of the ProbTransformer for property values of TPSA: 40.0; LogP: 2.0; SAS: 2.0 with an allowed deviation of 0.5, 0.1, and 0.1, respectively.

TPSA: 40.0, LogP: 2.0, SAS: 4.0

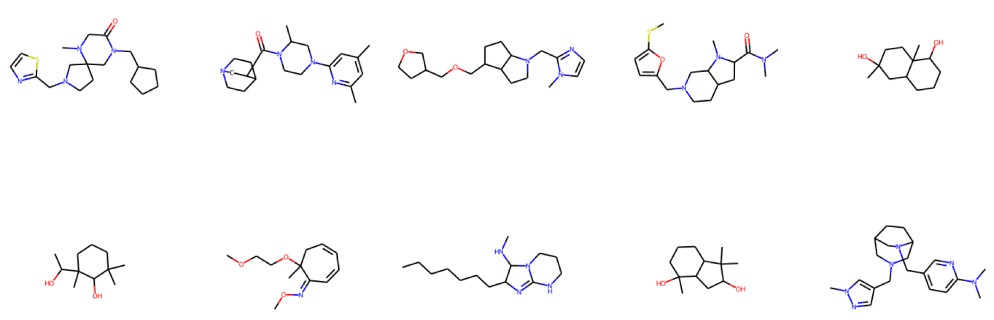

Figure 13: Example predictions of the ProbTransformer for property values of TPSA: 40.0; LogP: 2.0; SAS: 4.0 with an allowed deviation of 0.5, 0.1, and 0.1, respectively.

TPSA: 40.0, LogP: 6.0, SAS: 4.0

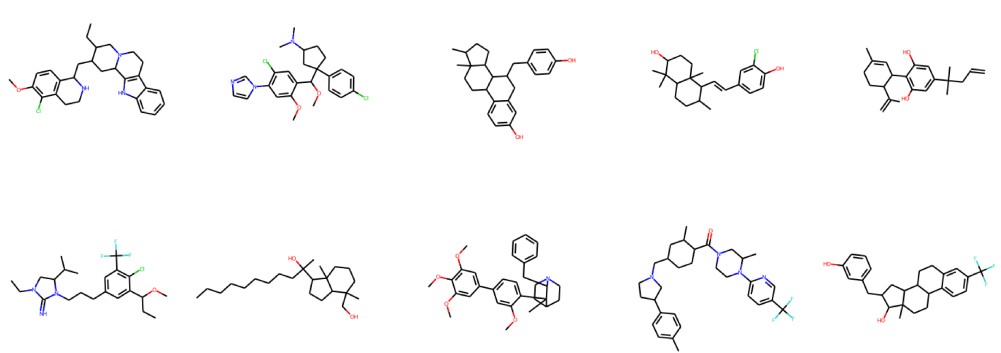

Figure 14: Example predictions of the ProbTransformer for property values of TPSA: 40.0; LogP: 6.0; SAS: 4.0 with an allowed deviation of 0.5, 0.1, and 0.1, respectively.

TPSA: 40.0, LogP: 6.0, SAS: 2.0

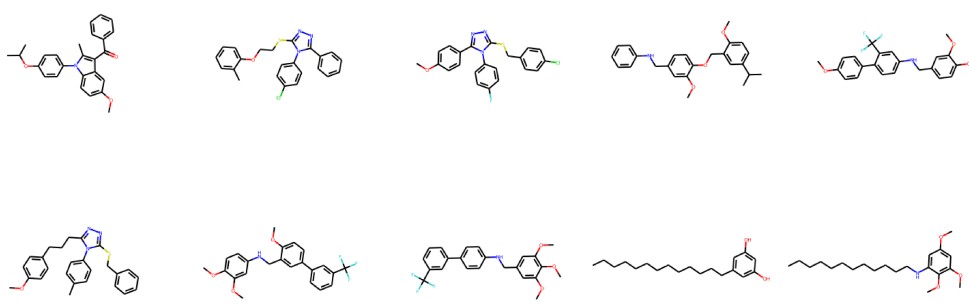

Figure 15: Example predictions of the ProbTransformer for property values of TPSA: 40.0; LogP: 6.0; SAS: 2.0 with an allowed deviation of 0.5, 0.1, and 0.1, respectively.

TPSA: 80.0, LogP: 2.0, SAS: 2.0

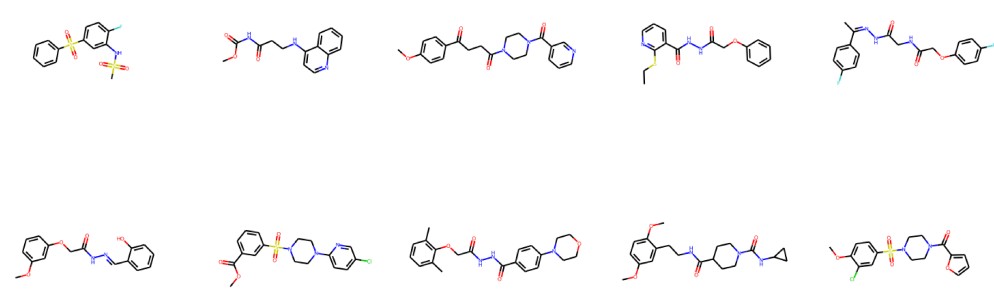

Figure 16: Example predictions of the ProbTransformer for property values of TPSA: 80.0; LogP: 2.0; SAS: 2.0 with an allowed deviation of 0.5, 0.1, and 0.1, respectively.

TPSA: 80.0, LogP: 2.0, SAS: 4.0

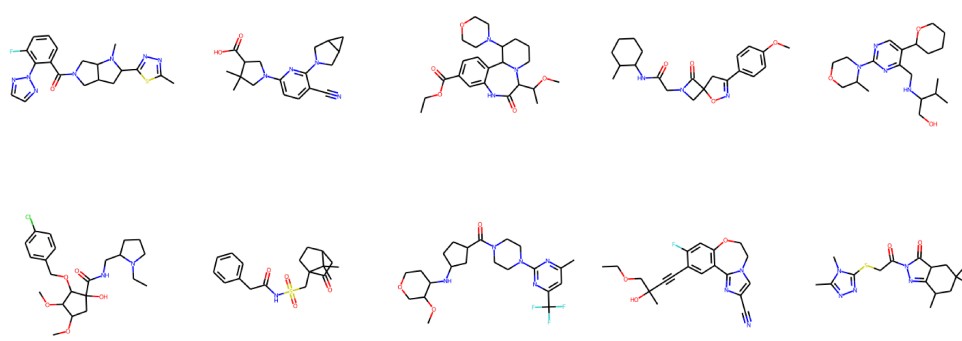

Figure 17: Example predictions of the ProbTransformer for property values of TPSA: 80.0; LogP: 2.0; SAS: 4.0 with an allowed deviation of 0.5, 0.1, and 0.1, respectively.

TPSA: 80.0, LogP: 6.0, SAS: 4.0

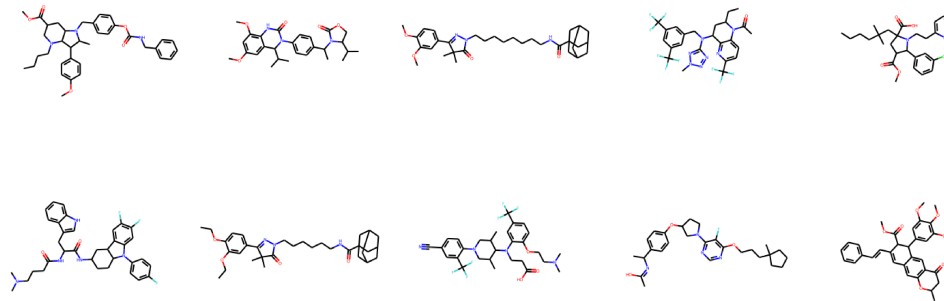

Figure 18: Example predictions of the ProbTransformer for property values of TPSA: 80.0; LogP: 6.0; SAS: 4.0 with an allowed deviation of 0.5, 0.1, and 0.1, respectively.

TPSA: 80.0, LogP: 6.0, SAS: 2.0

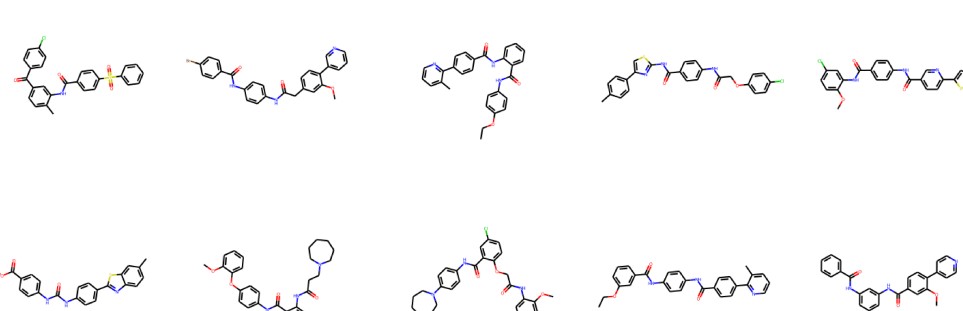

Figure 19: Example predictions of the ProbTransformer for property values of TPSA: 80.0; LogP: 6.0; SAS: 2.0 with an allowed deviation of 0.5, 0.1, and 0.1, respectively.

## D.4  Related Work

Inspired by progress in the field of natural language processing (NLP), early work employed recurrent neural networks (RNNs) to produce focus libraries based on the SMILES notation [73]. Later on, these approaches were coupled with reinforcement learning (RL) to focus the generation on molecules with desirable properties [69, 74]. Additional methods were proposed to tackle the problem, including generative adversarial networks (GANs) [70, 71], variational autoencoders (VAEs) [75, 76], and adversarial autoencoders (AAEs) [72, 77, 78, 79]. For more details on the different methods, we refer the interested reader to multiple reviews of the field [83, 84, 51, 53].

More recently, the success of self-attention mechanisms entered the field and novel methods were developed, adding attention either to RNNs [16] or VAEs [16, 118]. Remarkably, Transformer-based VAEs showed more complex latent representations of molecules and outperformed previous state-of-the-art VAEs [79] in the field [16].

However, as discussed before in Section 5, only some methods yet approached the challenging task of generating molecules with (multiple) predefined property values (conditional generation) [73, 80, 81, 82, 11].

# E Ablation Study

We list the hyperparameters of the Transformer and ProbTransformer as well as the training configuration for the ablation study in Table 21. We performed the training on one Nvidia RTX2080 GPU and the training time for one ProbTransformer model is ∼56h (1 prob layer) to ∼80h (all prob layer) and for one Transformer ∼33h. Also, we reduced the learning rate for the architecture ablation study due to unstable training when using all prob layers.

Table 21: Hyperparameters of the Transformer and ProbTransformer and training details of the ablation study.

| | |
|---|---:|
| Feed-forward dim | 2048 |
| Latent Z dim | 512 |
| Model dim | 512 |
| Number of layers | 6 |
| Number of heads | 8 |
| Prob layer | 4,3-4,2-5,all |
| Kappa | 0.1 |
| Dropout | 0.1 |
| Optimizer | adamW |
| Beta 1 | 0.9 |
| Beta 2 | 0.98 |
| Gradient Clipping | 100 |
| Learning rate schedule | cosine |
| Learning rate high | 0.0001/5 |
| Learning rate low | 0.00001/5 |
| Warmup epochs | 1 |
| Weight decay | 0.01 |
| Epochs | 200 |
| Training steps per epoch | 5000 |