# OpenReview forum: "Probabilistic Transformer: Modelling Ambiguities and Distributions for RNA Folding  and Molecule Design"
_NeurIPS.cc/2022/Conference — NeurIPS 2022 Accept_

### Official Review · Reviewer_J3YR · 2022-07-08

**Rating:** 5
**Confidence:** 3
**Soundness:** 3 good
**Presentation:** 3 good
**Contribution:** 3 good

**Summary:**

The authors present a probabilistic Transformer dubbed ProbTransfomer that models hierarchical latent distributions by integrating the transformer structure with a conditional variational auto-encoder. The article reveals that the incorporation of generalized ELBO with limited optimization is advantageous for variational training and that the proposed model excels at managing data ambiguities, including RNA folding and molecule design.


**Questions:**

1.	For the RNA folding problem, the authors may like to do a more comprehensive comparison of the ProbTransformer and other tools. For instance, independently evaluating the prediction of canonical and non-canonical base pairs.
2.	Considering adopting the ProbTransfomer model for predicting disordered protein areas. In contrast to the structural sections predicted by Alphafold2, disordered regions do not have fixed conformations and can therefore be seen as various target sequence conformational ensembles.
3.	Appendix line 754, the word “BRpRNA” should be changed to “BpRNA”.


**Strengths And Weaknesses:**

Strengths:
1.	The article is well organized and well written.
2.	Strong incentive exists where modeling unstructured molecules such as RNA is natural.
3.	Combining the Transformer with the cVAE, the authors propose a probabilistic layer for handling data ambiguity. ProbTransfomer's application to RNA folding and molecular design is a key development.
Weaknesses:
1.	The innovation is marginal as main part of proposed method is simply combination of existing works (Transformer (Ashish et al., 2017), cVAE (Kihyuk et al., 2015), ELBO (Simon et al., 2019), and GECO (Danilo et al., 2018).) Some recent advances(e.g. Grphformer) are absence in related work and discussions.
2. The authors introduce the ProbTransfomer model in an effort to solve the challenge of RNA folding. However, the likelihood of alternative RNA structures arising are mostly dictated by biological circumstances, which the authors do not examine. In addition, authors should consider RNA abundance when building datasets.

---

> ### Author Response · Authors · 2022-08-02
> **Individual response to Reviewer J3YR 2/2**
>
> > 2. The authors introduce the ProbTransfomer model in an effort to solve the challenge of RNA folding. However, the likelihood of alternative RNA structures arising are mostly dictated by biological circumstances, which the authors do not examine. In addition, authors should consider RNA abundance when building datasets.
>
> We would like to note that we did not build any new data sets for the RNA folding experiments ourselves, but rather build our data from commonly used data sets and splits that were defined in peer-reviewed work before, using the same protocols applied in these previous works (e.g. an 80% sequence similarity threshold between train and test data). We found a high level of ambiguity in these data sets regarding the sequences and the structures, and therefore decided to use RNA folding as an application to demonstrate the advantages of our model in learning in an ambiguous setting and to reconstruct the structure distributions of RNAs that existed in the data at hand. We agree with the reviewer that there are different ways of building data sets for RNA and that incorporation of biological circumstances to inform the models are desirable. However, we believe that these data sets have to be built elsewhere by experts in the respective fields. We are currently not aware of such comprehensive and publicly available data sets but would highly appreciate pointers to such data if the reviewer is aware of any.
>
> > For the RNA folding problem, the authors may like to do a more comprehensive comparison of the ProbTransformer and other tools. For instance, independently evaluating the prediction of canonical and non-canonical base pairs.
>
> We thank the reviewer for this suggestion and we included an evaluation performed on the test set TS0 in the Appendix C as Table 13.
>
> > Considering adopting the ProbTransfomer model for predicting disordered protein areas. In contrast to the structural sections predicted by Alphafold2, disordered regions do not have fixed conformations and can therefore be seen as various target sequence conformational ensembles.
>
> We thank the reviewer for pointing us to this very interesting field of application for our method and agree that the prediction of protein structure ensembles would be a suitable application that is in line with our current experimental setup and claims. That said, we already apply our work to hard problems in life sciences that, in our opinion, already support our claims very well (as also identified as a strength of our work by the reviewer), and a completely new additional application is not likely to be possible during the short rebuttal process. Nevertheless, we fully agree that it would be very interesting to apply our method to the prediction of protein structure ensembles and we will consider this as an application for future work.
>
>
> > Appendix line 754, the word “BRpRNA” should be changed to “BpRNA”.]
>
> We thank the reviewer for carefully reading our paper and we changed this minor issue.
>
> We would like to thank the reviewer again for their valuable comments and we are looking forward to an interesting discussion. We hope that our response clarified the remaining concerns of the reviewer and would like to kindly ask the reviewer to think about increasing our score.
>
> Best regards,
> the authors

---

> ### Author Response · Authors · 2022-08-02
> **Individual response to Reviewer J3YR 1/2**
>
> We thank the reviewer for the valuable feedback! We appreciate the receiver’s comments on the high quality of our manuscript and for pointing out that the application of our method to RNA folding and molecule design problems is a key development in the respective fields. In the following, we address the individual questions and concerns of the reviewer.
>
> > Weaknesses: 1. The innovation is marginal as main part of proposed method is simply combination of existing works (Transformer (Ashish et al., 2017), cVAE (Kihyuk et al., 2015), ELBO (Simon et al., 2019), and GECO (Danilo et al., 2018).) Some recent advances(e.g. Grphformer) are absence in related work and discussions.
>
> Novelty in science is often based on existing work. Even transformers can be seen as „just“ combining self-attention and feedforward layers in the right way, but have nevertheless been a breakthrough. Our work is indeed based on multiple existing components, but the knowledgeable combination is novel and leads to a novel system that improves the predictive performance in two important life science applications. Furthermore, the approach could be beneficial in many other applications with ambiguous data or the need for distribution learning on sequences. Also, we differ from a conventional cVAE in the sense that we model and learn a hierarchical probabilistic latent space. We further introduce the novel kappa annealing, which stabilizes our training and could be beneficial for variational training in general.
>
> We thank the reviewer for pointing us to the Graphformer. The Graphformer takes as input an existing graph while we use a sequence of nucleotides as input and create a dot-bracket notation and an adjacency matrix.

---

### Official Review · Reviewer_GgjC · 2022-07-10

**Rating:** 7
**Confidence:** 3
**Soundness:** 3 good
**Presentation:** 3 good
**Contribution:** 3 good

**Summary:**

The paper introduces a novel probabilistic layer into the transformer architecture that incorporates cVAE, which allows multiple samples and predictions that as an ensemble may correspond to some distribution of interest. It also introduces a training strategy ("kappa annealing") to effectively train models with such layers while reducing need for hypers tuning.

**Questions:**

See above.

**Limitations:**

Adequately addressed.

**Strengths And Weaknesses:**

While the elements of the main ideas are present, they are presented in a mathematically unclear manner with an unconvincing synthetic experiment (section 4.1) or experiments (section 4.2 and 4.3) whose significance is not well described for readers who are not familiar with the exact metrics used.

Examples of imprecise language and notations:
- Abstract: "Therefore, we propose a hierarchical latent distribution to enhance one of the most successful deep learning models, the Transformer, to accommodate ambiguities and data distributions." What do the authors mean by "ambiguities and data distributions" mathematically and how do the chosen experiments address these?
- L76-75: "This objective at training time can be viewed as a reconstruction task, which is an easier task than prediction." Why is this easier?
- L99-101: "As a result of this hierar-chical composition of the ProbTransformer, the cVAE’s (conditional) prior model pθ (zm |Z<m , X )and generation model pρ(Y |X, Z) cannot be disentangled cleanly anymore in it and effectivelybecome a single model pΦ(Y |X)." What precisely do you mean by "disentangled cleanly anymore" and how does this relate to the training strategy?
- Eq4-6: what is i?
- Notations in Section 3.2 is under-defined and the material could be better presented for more effective communication. For example, show how the loss terms can be derived in a more principled manner from an optimization objective and variational approximation; what are p_m and q_m in Eq 9? A read could possibly infer from the context but it is on the authors to make this explicit.

Suggestions for experiments
- 4.1. I suggest to show some examples in the main text and how the tasks, objectives, and metrics are appropriate for quantifying the model's capability for "ambiguities and data distributions"; otherwise, readers can only guess what the authors mean. This synthetic example feels contrived and is not intuitive and convincing of how the proposed addition to the transformer architecture provides a meaningful advantage.

4.2. and 4.3. Similar to above, I suggest to show some examples in the main text and how the tasks, objectives, and metrics are appropriate for quantifying the model's capability for "ambiguities and data distributions"; otherwise, readers can only guess what the authors mean.

---

> ### Author Response · Authors · 2022-08-02
> **Individual response to Reviewer GgjC 5/5**
>
> > 4.1. I suggest to show some examples in the main text and how the tasks, objectives, and metrics are appropriate for quantifying the model's capability for "ambiguities and data distributions"; otherwise, readers can only guess what the authors mean. This synthetic example feels contrived and is not intuitive and convincing of how the proposed addition to the transformer architecture provides a meaningful advantage.
>
> We include a detailed description of the synthetic task in the appendix of our work. We further would like to note that the synthetic example is constructed to evaluate the capabilities of generative models to learn hidden distributions of the data but that it does *NOT* favor our proposed method in any way. We believe that the empirical results in this section indicate that our method benefits from learning a hierarchical latent distribution that seems to capture the underlying hidden distributions in the data better than a vanilla transformer does.
>
>
> > 4.2. and 4.3. Similar to above, I suggest to show some examples in the main text and how the tasks, objectives, and metrics are appropriate for quantifying the model's capability for "ambiguities and data distributions"; otherwise, readers can only guess what the authors mean.
>
> We explain all experiments in detail in the appendix of our paper. The metrics we use are commonly used by other approaches and previous work in the respective fields (for 4.2 we use the additional metrics “Hamming distance” and “number of solved tasks” to have a better measure of performance because there are known issues with the commonly used F1-score regarding conformational changes of RNAs). However, we agree with the reviewer that the RNA folding task should be explained in more detail in the main text and we updated our manuscript accordingly.
>
> We would like to thank the reviewer again for the constructive criticism. However, while we think that many of the concerns and questions of the reviewer are reasonable, some of them probably result from misunderstanding parts of our manuscript, and we hope that we could clarify these parts as well as all concerns and questions of the reviewer with our response. We therefore would like to kindly ask the reviewer to increase their score of our work. We are looking forward to a fruitful discussion phase and welcome any further questions or suggestions from the reviewer.
>
> Best regards,
> the authors

---

> > ### Comment · Reviewer_GgjC · 2022-08-02
> > **comment**
> >
> > Thank you for your constructive, detailed responses. Again, with the clarifications, the revised manuscript read a lot clearer and I have updated my review and scores accordingly.

---

> > > ### Author Response · Authors · 2022-08-03
> > > **Final response to Reviewer GgjC**
> > >
> > > Dear reviewer GgjC,
> > >
> > > We thank you very much for the fast reply and for addressing all our comments.
> > > We are glad that we could clarify your questions and concerns with our response. We updated the abstract according to the points of discussion. We appreciate that you changed the review and increased our scores in response to our comments and proposed improvements. Generally, we thank you for being fair and kind throughout the entire discussion and for acknowledging the additional contributions of our work.
> > >
> > > Best regards,
> > > the authors

---

> ### Author Response · Authors · 2022-08-02
> **Individual response to Reviewer GgjC 4/5**
>
> > Eq4-6: what is i?
>
> We again thank the reviewer for bringing up any concerns. However, there is no variable ‘i’ in equations 4-6. We speculate that the reviewer is confused by the lowercase ‘in’ (equations 4 and 5) and in response changed the lowercase notations of both, ‘in’ and ‘out’, to uppercase ‘In’ and ‘Out’ in equations 4, 5, and 7 to avoid confusion of future readers.
>
> > Notations in Section 3.2 is under-defined and the material could be better presented for more effective communication. For example, show how the loss terms can be derived in a more principled manner from an optimization objective and variational approximation; what are p_m and q_m in Eq 9? A read could possibly infer from the context but it is on the authors to make this explicit.
>
> We would like to make our manuscript as strong as possible and are very willing to react to any reasonable criticism by incorporating changes into our paper. However, the reviewer is pointing to an entire section of our work where a more detailed explanation of missing definitions and undefined variables would have been more helpful to improve our manuscript. We will therefore address the two explicit questions of the reviewer here and review the section again for any undefined variables and clean the notation as follows:
>
> * We introduce the implicitly defined distribution P as the prior distribution in lines 133-134 to explicitly point out that this distribution is the distribution of the generating model. In particular, we would like to change lines 133-134 to the following. Would the reviewer agree that this clarifies the concerns?
>
> The negative ELBO loss $\mathcal{L}_{ELBO}$ (Equation 1) is composed of a reconstruction loss $\mathcal{L}_{rec}$ and a Kullback-Leibler divergence D_{KL} between the prior distribution P conditioned on the latent Z^{post} drawn from Q, and the posterior distribution Q.
>
> We note, however, that the sequential relation between the different z’s ~ P of the prob layers in the generating model do not only come from the generating model itself but are a result of the inference when drawing the z’s from Z_{<m} from the posterior model Q at each layer m, as becomes obvious in equation 9. However, we think that this is the most straightforward and understandable way of introducing P explicitly.
>
> Regarding question 1 [show how the loss terms can be derived in a more principled manner from an optimization objective and variational approximation;]:
>
> The loss terms we use are not novel and build upon existing work. We do not agree with the reviewer that we should repeat the derivation of a formula that has already been shown in prominent previous work. That said, we would like to emphasize that there is still a novelty in applying the GECO objective in the given transformer setting, and we additionally provide empirical evidence that our proposed novel annealing strategy, kappa annealing, could be beneficial for variational training in general.
>
> Regarding question 2 [what are p_m and q_m in Eq 9]:
>
> We agree with the reviewer that Equation 9 is slightly under-defined with regard to q_m and p_m. We, therefore, now rephrased line 137 to introduce the formula as follows:
>
> and D_{KL} is the sum of KL divergences between the hierarchical decompositions, p_m and q_m, of the distributions P and Q:
>
> Does this clarify our text to the reviewer?

---

> > ### Comment · Reviewer_GgjC · 2022-08-02
> > **comment**
> >
> > Thank you for your response.
> >
> > "The loss terms we use are not novel and build upon existing work. We do not agree with the reviewer that we should repeat the derivation of a formula that has already been shown in prominent previous work." that's a fair point. i recognize that this is a matter of personal preference than a strong requirement for clarity.
> >
> > re "... to change lines 133-134 to the following. Would the reviewer agree that this clarifies the concerns?" yes.
> >
> > "...and D_{KL} is the sum of KL divergences between the hierarchical decompositions, p_m and q_m, of the distributions P and Q. Does this clarify our text to the reviewer?" yes.

---

> ### Author Response · Authors · 2022-08-02
> **Individual response to Reviewer GgjC 3/5**
>
> > L76-75: "This objective at training time can be viewed as a reconstruction task, which is an easier task than prediction." Why is this easier?
>
> We thank the reviewer for asking this important question and agree that this sentence might be unclear to readers that are not familiar with the field but also agree that this part is important for understanding the remainder of the paper. For clarification:
>
> The training and testing phases of cVAE differ mainly in the drawing of latent variables z to inform the generation network’s predictions. While at test time, these variables are drawn from the prior model p_\theta(z|x), z’s are drawn from the recognition network q_\Phi(z|x, y) at training time. Since the recognition network is informed by the true label y, the task during training can be viewed as a reconstruction task, which makes the learning of this process easier than a pure prediction task.
>
> In a reconstruction task, the expected output of the model, “the prediction”, is the same as the given input. Therefore all required information is available to the model and it can, in principle, learn the perfect prediction (CE loss = 0). In contrast, a prediction task maps from a given input to a prediction defined by a label. But it is often unclear if the input contains all information to create a perfect prediction. This makes a reconstruction task easier to handle than a prediction task. By knowing that we can theoretically achieve (CE loss = 0), we can adjust an information bottleneck or auxiliary loss (like in our case the KL-loss).
>
>
> > L99-101: "As a result of this hierarchical composition of the ProbTransformer, the cVAE’s (conditional) prior model pθ (zm |Z<m , X )and generation model pρ(Y |X, Z) cannot be disentangled cleanly anymore in it and effectively become a single model pΦ(Y |X)." What precisely do you mean by "disentangled cleanly anymore" and how does this relate to the training strategy?
>
> We agree with the reviewer that the phrasing of this part of our paper is unfortunate and thank the reviewer for pointing this out. For clarification:
>
> As stated in the sentence before the one mentioned by the reviewer, a sequential relation between the latent variables z of each layer is achieved by the attention mechanism in each block. This is in stark contrast to the distributions modeled in the cVAE which do not relate the z’s sequentially. As a result, the prior model p_\theta(z|x) (or better each p_\theta(z_m|z<m, x)) and the generating model of the ProbTransformer cannot be viewed as two distinct models anymore but effectively become a single model p_\Phi(Y|X). Since the posterior network that is used during training of the ProbTransformer has the same architecture as the generating network, this effect can also be observed during training, where the attention mechanism relates the latent variables z_1, …, z_m sequentially in both models. Note that both models are trained jointly and only the weighting of the reconstruction vs. the KL loss is affected by the training strategy (GECO instead of ELBO). We now changed this part in the manuscript as follows:
>
> As a result of this hierarchical composition of the ProbTransformer and this sequential relation, the prior model pθ (zm |Z<m , X ) and the generation model pρ(Y |X, Z) effectively become a single model pΦ(Y |X), which differs from the modeling strategy of the cVAE.

---

> > ### Comment · Reviewer_GgjC · 2022-08-02
> > **comment**
> >
> > Thank you for your response. I found the changes clarify.

---

> ### Author Response · Authors · 2022-08-02
> **Individual response to Reviewer GgjC 2/5**
>
> > While the elements of the main ideas are present, they are presented in a mathematically unclear manner
>
> We are surprised about the reviewers' concerns regarding our mathematical presentation in the paper, which we believe is in line with previous work in the field. Nevertheless, we would like to thank the reviewer for sharing these concerns and for giving us the chance to review our mathematical descriptions and formulas to avoid confusing other readers that aren’t familiar with the topic and the fields of applications. We would therefore like to kindly ask the reviewer to specify the unclear parts.
>
>
> > with an unconvincing synthetic experiment (section 4.1) or experiments (section 4.2 and 4.3) whose significance is not well described for readers who are not familiar with the exact metrics used.
>
> The motivation for the synthetic task is to create a task that, on the one hand, is challenging, and on the other hand, is a task that allows access to the true data distribution. We described the synthetic distribution task in detail in Appendix B and refer to this section from the main paper.
>
> We agree with the reviewer that the main body lacks a formal description of the RNA folding problem. We changed our manuscript accordingly and a general definition of RNA folding can now be found in the beginning of section 4.2. The task of conditional generation of molecules is introduced in Section 4.3 and the general setup can be found in the setup paragraph. Both tasks and the metrics are explained in detail in Appendices C and D, respectively.
>
> > Abstract: "Therefore, we propose a hierarchical latent distribution to enhance one of the most successful deep learning models, the Transformer, to accommodate ambiguities and data distributions." What do the authors mean by "ambiguities and data distributions" mathematically and how do the chosen experiments address these?
>
> We agree with the reviewer that a certain formalism, including mathematical descriptions and formulas, could often be necessary and helpful to understanding very technical parts of a paper. However, we also think that such formalisms should be used as sparsely as possible and only if really required to deepen the understanding. Regarding our abstract, we slightly disagree with the reviewer that such a formalism is required at this point in our work. Nevertheless, we changed our abstract as follows to point out the meaning of ambiguities and distributions:
>
> Our world is ambiguous and this is reflected in the data we use to train our algorithms. This is especially true when we try to model natural processes where collected data is affected by noisy measurements and differences in measurement techniques. Sometimes, the process itself can be ambiguous, such as in the case of RNA folding, where a single nucleotide sequence can fold into multiple structures **that occur with different probabilities**. This ambiguity suggests that a predictive model should have similar probabilistic characteristics to match the data it models. Therefore, we propose a hierarchical latent distribution to enhance one of the most successful deep learning models, the Transformer, to accommodate **these sorts of** ambiguities and data distributions. We show the benefits of our approach **by learning the hidden distribution** on a synthetic task, with state-of-the-art results in RNA folding **when training on highly ambiguous data, capable of reconstructing structure distributions** and demonstrate its generative capabilities on property-based molecule design **by implicitly learning the underlying property distributions and** outperforming existing work.
>
> Would these changes clarify the meaning of ambiguities and distributions to the reviewer and how our experiments relate to these two?
> We haven't changed it yet since we consider the change of the abstract as a major change and would like to discuss it beforehand.

---

> > ### Comment · Reviewer_GgjC · 2022-08-02
> > **Comment**
> >
> > Thank you for your response. I find helpful the suggested changes to the abstract and also acknowledge that my overall comment could have been more specific. I found that the revised versions was clearer and read more easily.

---

> ### Author Response · Authors · 2022-08-02
> **Individual response to Reviewer GgjC 1/5**
>
> We thank the reviewer for the constructive criticism and suggestions for improving our work. However, we would like to clarify some points of confusion and misunderstanding that hopefully lead to improving the reviewer's view and rating of our work. We will detail all points in the following.
>
> > Summary:
> The paper introduces a novel probabilistic layer into the transformer architecture that incorporates cVAE, which allows multiple samples and predictions that as an ensemble may correspond to some distribution of interest. It also introduces a training strategy ("kappa annealing") to effectively train models with such layers while reducing need for hypers tuning.
>
> We thank the reviewer for this summary, however, we do not think that it correctly captures our work in its entirety. We would disagree that the probabilistic sampling behaves “as an ensemble” because of the hierarchical probabilistic latent space. Sampling from the first layer affects further sampling in higher layers. Further, the training objective is GECO, and kappa annealing is a technique to automatically adapt the kappa hyperparameter. On the one hand, this reduces the need for optimization of this parameter and on the other hand, it stabilizes the training due to an incremental increase of the reconstruction difficulty by decreasing the kappa value. Our novel annealing strategy further adds value to the community as it could stabilize variational training in general.
>
> Additionally, we would like to point out that our probabilistic layer could be modularly used in different generative models, and that we achieve state-of-the-art performance across a wide range of different tasks. In this regard, we would like to highlight the learning and reconstruction of structure distributions of RNA sequences, which, to our knowledge, describes the first approach capable of learning these distributions from data and can be considered a key step in the field.

---

> > ### Comment · Reviewer_GgjC · 2022-08-02
> > **comment**
> >
> > Thank you for the additional clarification. I agree with the authors' comments.

---

### Official Review · Reviewer_Ex3E · 2022-07-12

**Rating:** 8
**Confidence:** 3
**Soundness:** 4 excellent
**Presentation:** 3 good
**Contribution:** 4 excellent

**Summary:**

This paper introduced ProbTransformer, which is a transformer based generative model. By adding a probabilistic architecture to the original transformer, ProbTransformer was able to model the hidden distribution of the data and provided probabilistic predictions. In addition, its contributions also include the online adaptation technique, kappa annealing, to gain better stability and performance. This paper is driven by the challenge of modeling the ambiguities of certain tasks, like the RNA folding problem. Deterministic models may harm the performance on those problems when one sample can have ambiguous outputs. The manuscript empirically demonstrates that the addition of their probabilistic layer improves not only the ability to model the latent data distribution but the stability of the learning process combined with the kappa annealing.


**Questions:**

see above

**Limitations:**

Yes

**Strengths And Weaknesses:**

Strengths:
==========

The paper addresses an important question of modeling uncertainty in structure modeling. They show their modeling approach is applicable to different problems/domains.

The flow of this paper is clear. It introduces the motivation first and then its solution to the problem fits the motivation nicely. In addition, the experimental design supports their claims by showing their model outperforming the state-of-art models in different downstrain tasks. Specifically:

The authors nicely demonstrate improvements in the quality of distribution learning by using the ProbTransformer in the synthetic sequential dataset.

To enhance the claim of distribution learning, the paper further shows that it can succeed in both RNA folding and molecule design problems, which use the encoder-only model and the decoder-only model to solve respectively. In both tasks, the authors provide some examples to illustrate their good performance while their evaluation metrics show that their model outperforms the other models in general.

Using ablation studies, the authors demonstrate the effect and also the importance of the hierarchical probabilistic and kappa annealing design.


Weaknesses:
==============
The paper did not illustrate the model design very clearly. This is not a minor comment but an important one given what the authors are trying to achieve/claim here. The paper’s annotations and formulas give the audience a basic understanding of what the authors aim to do. However, it is hard to follow how the model is constructed. They should have either in the main text or supplementary a “walk through” example to clearly illustrate the model’s working. It is vague about the definition of encoder-only and decoder only models and how they are built differently in the downstream tasks.

The authors mention that the encoder-decoder can not be built for now because of the same length issue, but it needs to be clarified why the model can’t make the same length for the output of the encoder with the input of the decoder. This relates to the previous comment as well.

The authors state RNA structure can be expressed by a dot-bracket notation, which means source and target can have the same length. However, do-bracket only captures nested dependencies and is not able to capture structures such as pseudo-knots. This is a key issue with the model which the authors sweep under the carpet. In contrast, other models are not limited by this (some none nested structures are shown in the predicted structures they included). Some discussion of these limitations compared to other methods is warranted, and evaluation would have been nice though possibly out of scope.

Previous work also showed that RNA structure prediction methods greatly depend on how many canonical nested base pairing occurs - the fewer the ratio, the worse the model.  Again, at least mentioning/discussion of limitations is important.

To justify the model in the content of RNA the authors seem to mix a bit claims about mutations and molecules that shift between different structures in specific regions. It’s not clear to these reviewers the data actually support claims there. The authors should clarify/clean the text around these statements and make sure claims and results are well connected.

From the description in the main text it seems that the authors took a different approach compared to the authors of MXFold2 and SPOT RNA when defining train/test because they wanted to retain ambiguous structures. It is not clear if the comparisons done in the paper allowed these models to be trained in the same way. If not, there is the doubt that improvements are due at least to some extent to differences in training.

Minor comments:
=============
Line 21: “Vice-versa” is unclear in this context.
Line 169: insights in the
Line 174-175: “token…is a unique distribution”
Lack of labels for different kappa values in Figure 4.

---

> ### Author Response · Authors · 2022-08-02
> **Individual response to reviewer Ex3E 3/3**
>
> > From the description in the main text it seems that the authors took a different approach compared to the authors of MXFold2 and SPOT RNA when defining train/test because they wanted to retain ambiguous structures. It is not clear if the comparisons done in the paper allowed these models to be trained in the same way. If not, there is the doubt that improvements are due at least to some extent to differences in training.
>
> We thank the reviewer for carefully reading our manuscript.
> We first would like to point out that all data we are using in this work was also available for training the methods we compare to. We use the exact same validation and test splits as described in SPOT-RNA (which were also used for evaluation of MXFold2). In line with this work, we removed samples from the training set that show sequence similarity greater than 80% with any of the sequences in (any) of the test and validation sets using the same tool, CD-HIT.
> However, for the training of SPOT-RNA, the authors further removed sequences from the training data at an 80% similarity cutoff within the train set, argumenting with avoidance of overfitting. We do not clearly agree with this argument of overfitting here, which should already be implicitly captured by having removed sequences from the train set that show similarity with the test samples but do not want to speculate about further reasons for removing ambiguous data to potentially balance the training set. We think that an RNA folding algorithm should work out-of-the-box for any input sequence and that all methods reasonably use the best data they can to train their algorithms in order to achieve this goal, probably including the removal of ambiguous data from the train set. For training of our method, we skipped the step of additionally removing ambiguous samples from the training data, since our model should be strong at learning in this ambiguous setting (as also shown by our results). The training pipeline of SPOT-RNA, our strongest competitor, is undisclosed and training of MXFold2 would require data preprocessing steps since MXFold2 cannot predict all base pairs nor pseudoknots. We, therefore, did not re-train the competitors on the more ambiguous train set but we would also not expect a strong improvement in performance. We again emphasize that the test sets used are identical.
>
> > Line 21: “Vice-versa” is unclear in this context.
> > Line 169: insights in the
> > Line 174-175: “token…is a unique distribution”
> > Lack of labels for different kappa values in Figure 4.
>
> We would like to again thank the reviewer for carefully reading our paper. We addressed these minor issues in the manuscript.
> We hope that we can assure the reviewer of the strength of our work with this reply and would like to kindly ask the reviewer to champion our work. We are looking forward to any further questions about our paper or rebuttal in the discussion phase.
>
> Best regards,
> the authors

---

> > ### Comment · Reviewer_Ex3E · 2022-08-08
> > **Reply  Ex3E 3/3**
> >
> > OK so as I assumed the train/test splits are the same but you didn't retrain the competing models without removal of redundancy in the train. As we noted, it is still possible that could have been an issue of the original training procedure (rather than the actual model) but I understand it's too much of a burden to re train those models just for checking this point. Just be sure to make this difference clear in the final version.

---

> > > ### Author Response · Authors · 2022-08-09
> > > **Response to Reviewer Ex3E**
> > >
> > > We thank the reviewer for this reply. We changed our manuscript accordingly now and will keep the changes for the final version.
> > > We did the following changes to our manuscript
> > > - We add a detailed figure of the model in the Appendix
> > > - We remove all statements that refer to mutations and explain in more detail that we mean differences in the sequence that sometimes might result in the same secondary structure to clarify this claim.
> > > - We clarified that we did not remove ambiguous sequences from the training data which is in contrast to previous work.
> > >
> > > Best regards,
> > > the authors

---

> ### Author Response · Authors · 2022-08-02
> **Individual response to reviewer Ex3E 2/3**
>
> > The authors state RNA structure can be expressed by a dot-bracket notation, which means source and target can have the same length. However, do-bracket only captures nested dependencies and is not able to capture structures such as pseudo-knots. This is a key issue with the model which the authors sweep under the carpet. In contrast, other models are not limited by this (some none nested structures are shown in the predicted structures they included). Some discussion of these limitations compared to other methods is warranted, and evaluation would have been nice though possibly out of scope.
>
> We thank the reviewer for pointing us to this issue and we think we should briefly clarify this here. First of all, we would like to point out that our model is *NOT* limited to the prediction of nested RNA structures only. We, however, understand the confusion of the reviewer due to our unfortunate usage of the term ‘dot-bracket notation’. In fact, our model can predict non-nested (pseudoknotted) RNA structures up to a page number of 3 (see https://academic.oup.com/nar/article/46/11/5381/4994207; Figure 1F for an illustration of the page number) using the extended dot-bracket notation as e.g. described here:
> https://www.tbi.univie.ac.at/RNA/ViennaRNA/doc/html/rna_structure_notations.html
> We restrict our predictions to strings in extended dot-bracket notation using only additional brackets for the description of pseudoknots due to a lack of data with higher-order pseudoknots above page number 3. We, therefore, did not use the term ‘extended dot-bracket notation’, because we currently do not support the entire repertoire of page numbers that could be described in the extended dot-bracket notation. We note, that this is a design choice we made due to a lack of available data and that it is *NOT* a general limitation of the proposed model.
> As already stated by the reviewer, we also show examples of predictions of our model that contain non-nested base pairs, demonstrating its ability to predict these base pairs correctly (see Figure 9 in the appendix). Further, the example predictions in the main paper (Figure 3) highlight that, in contrast to SPOT-RNA, our model typically does not predict pseudoknots for structures that do not contain pseudoknots.
> Finally, we would like to add that we discuss this issue very briefly in a footnote in line 843 in the appendix and that we further discuss the limitation of RNAFold and MXFold2 of being capable of predicting nested RNA structures only in lines 929-930 (both in the Appendix provided with the supplementary material).
> That said, we agree with the reviewer on this issue and understand the point of confusion. We now:
> * clarify the scope of predictions in the main paper and
> * add a discussion on pseudoknot predictions.
>
> > Previous work also showed that RNA structure prediction methods greatly depend on how many canonical nested base pairing occurs - the fewer the ratio, the worse the model. Again, at least mentioning/discussion of limitations is important.
>
> We thank the reviewer for pointing us to this very interesting correlation. Since we were not aware of this fact before, we would like to kindly ask the reviewer to provide a citation to include such analysis in future work.
>
> > To justify the model in the content of RNA the authors seem to mix a bit claims about mutations and molecules that shift between different structures in specific regions. It’s not clear to these reviewers the data actually support claims there. The authors should clarify/clean the text around these statements and make sure claims and results are well connected.
>
> We agree with the reviewer that this part of our manuscript bears some potential for confusion due to very dense writing. For clarification:
>
> For our experiments on RNA folding, we consider three settings: (1) A general evaluation on the test set TS0 to compare the capabilities of our method to learn the folding process to existing work, (2) an evaluation on different sequences that share the same secondary structure, and (3) an evaluation on sequences where we had multiple annotated secondary structures available.
> While experiment (3) clearly addresses the prediction of structure ensembles and the learning of their underlying distribution (or the shift between different structures), experiment (2) evaluates the learning of ambiguities in the sequence data. Such data could, e.g., be a result of a mutation in the real world and we, therefore, think that our motivation, including the statement on mutation, is correct. However, we agree with the reviewer that we did not include more detailed analysis on this but left the statement open for interpretation. We addressed this by:
>
> * Carefully check this statement again to not make any undeserved claims and disentangle the two sentences to clarify that we are speaking about the structure space in the first and about the sequence space in the second.

---

> > ### Comment · Reviewer_Ex3E · 2022-08-08
> > **Reply to Ex3E 2/3**
> >
> > "We thank the reviewer for pointing us to this very interesting correlation. Since we were not aware of this fact before, we would like to kindly ask the reviewer to provide a citation to include such analysis in future work."
> >
> > See 10.1073/pnas.2112677119 Fig5
> >
> > "(2) an evaluation on different sequences that share the same secondary structure"
> > Are these actually mutated sequences or shared structures in say a family of RNA? Maybe make your data/claims more precise here. I would argue that these are not exactly the same kind of samples/problems. The mutations are not "naturally occurring" and not directed by natural selection. Predicting their effect is somewhat of a different (and important) task (e.g. for disease).
> >
> > Also it is not easy to asses the changes in your write up - it makes the job of the reviewers, dealing with multiple papers, much easier if you upload a version with all the changes to the text/figures marked clearly.

---

> > > ### Author Response · Authors · 2022-08-09
> > > **Response to Reviewer Ex3E**
> > >
> > > >See 10.1073/pnas.2112677119 Fig5
> > >
> > > We appreciate the detailed link.
> > >
> > > >"(2) an evaluation on different sequences that share the same secondary structure" Are these actually mutated sequences or shared structures in say a family of RNA? Maybe make your data/claims more precise here. I would argue that these are not exactly the same kind of samples/problems. The mutations are not "naturally occurring" and not directed by natural selection. Predicting their effect is somewhat of a different (and important) task (e.g. for disease).
> > >
> > > While we think that, given the right set of data, our probabilistic transformer would be capable of capturing differences in a mutation regimen, we agree with the reviewer that our data is probably not providing enough evidence for such claims. Since we do not want to make misleading claims with the term ‘mutation’ here, but we still think we capture an important problem when learning from sequence ambiguities, we changed the part of the paper accordingly.
> > >
> > > >Also it is not easy to asses the changes in your write up - it makes the job of the reviewers, dealing with multiple papers, much easier if you upload a version with all the changes to the text/figures marked clearly.
> > >
> > > We appreciate the work of the reviewers and understand this reply. We will at the end of our response provide a comprehensive list of the new changes.

---

> ### Author Response · Authors · 2022-08-02
> **Individual response to reviewer Ex3E 1/3**
>
> We thank the reviewer for the comprehensive review, and for pointing out the importance of our work in addressing the problem of modeling uncertainty, ambiguities, and distributions. We further appreciate the comments on the organization, structure, and flow of our manuscript and the setup of the experiments to support our claims as well as pointing out the novelty of our work to stabilize variational training and the improvements in performance that we achieve across a wide range of domains.
>
> We also appreciate the questions and concerns of the reviewer that will help us to further improve our work. We address these questions in the following with details on how we will change or changed our manuscript in response to these points of discussion.
>
> > The paper did not illustrate the model design very clearly. This is not a minor comment but an important one given what the authors are trying to achieve/claim here. The paper’s annotations and formulas give the audience a basic understanding of what the authors aim to do. However, it is hard to follow how the model is constructed. They should have either in the main text or supplementary a “walk through” example to clearly illustrate the model’s working.
>
> To address this issue, we will include a more detailed and multi-step figure based on Figure 2 in the supplementary material. Would you also like us to add a more detailed introduction to the Transformer architecture in general?
>
> > It is vague about the definition of encoder-only and decoder only models and how they are built differently in the downstream tasks.; The authors mention that the encoder-decoder can not be built for now because of the same length issue, but it needs to be clarified why the model can’t make the same length for the output of the encoder with the input of the decoder. This relates to the previous comment as well.
>
> We thank the reviewer for this clarification question. The benefit of an encoder-decoder architecture is that the source and target sequences don’t need to have the same length. Otherwise, an encoder-only model is more efficient to train. But our training setup requires the source and target sequence as input to the posterior model. If they have different lengths we run into alignment issues. We observed in preliminary experiments on neural machine translation that this harms the overall performance. We clarify this point in more detail in our paper now [lines 85-86:
>
> It also applies to the decoder-only model but not directly to an encoder-decoder model because our training setup requires the alignment of the source and target sequences in the posterior input.

---

> > ### Comment · Reviewer_Ex3E · 2022-08-08
> > **Reply to Ex3E 1/3**
> >
> > "include a more detailed and multi-step figure based on Figure 2 in the supplementary material. Would you also like us to add a more detailed introduction to the Transformer architecture in general?"
> >
> > I do not think a Transformer intro is necessary - this is NeurIPS so it's reasonable to assume Transformers are generally known, but the specifics should be illustrated.
> >
> > I didn't see such a figure in the supplementary in the latest version you uploaded so I can assess it. Also the supplementary file is very "heavy" and both Acrobat Reader and Firefox struggle with it on my computer. I suspect its your Fig11 in vector format with all the dots - try saving it first as a PNG at high res.

---

> > > ### Author Response · Authors · 2022-08-09
> > > **Response to Reviewer Ex3E**
> > >
> > > We agree with the reviewer that a Transformer intro might not be necessary. The reviewer is correct in that we did not include a more detailed figure yet before making sure what precisely the reviewer was missing in our work. We now included two new illustrations of the inference and training in Appendix A.
> > >
> > > We also thank the reviewer for this helpful comment on our supplementary material and changed the figure to PNG format.

---

### Author Response · Authors · 2022-08-02
**General response**

Dear reviewers,

We would like to thank all reviewers for their valuable feedback! In particular, we appreciate the comments on the importance of the question we address and the relevance of improving the quality of distribution learning, the clear structure of our experiments to support our claims, as well as the positive comments regarding the organization and flow of our paper. Furthermore, we thank you for pointing out that the application of our work to the problem of RNA folding and molecular design can be viewed as a key development in these fields. In order to keep the overhead for each reviewer as low as possible, we address all questions and concerns of the reviews in separate responses to the individual reviews. We are looking forward to an inspiring discussion phase next week.

Best regards,
the authors.

---

### Meta-Review · Area_Chair_3Mc5 · 2022-08-26

**Recommendation:** Accept
**Confidence:** Certain

**Metareview:**

This is a nice application-motivated paper that introduces and tests a novel stochastic variant of a transformer architecture.

All three reviewers recommend acceptance (albeit one is borderline). The borderline review focuses on the closeness of this to existing work, and the relatively incremental nature of the contribution. While I do see the potential concern, I think all in all the consensus is clearly to accept.

Interaction between reviewers and authors led to a number of beneficial changes to the paper during the review process.

**Award:**

No

---

### Decision · Program_Chairs · 2022-09-14

Accept